# Follow the Energy, Find the Path: Riemannian Metrics from Energy-Based Models

**Louis Bethune***
Apple

**David Vigouroux**
IRT Saint Exupéry, ANITI, IMT Atlantique

**Yilun Du**
Harvard University

**Rufin VanRullen**
CNRS

**Thomas Serre**
Brown University

**Victor Boutin***
CNRS

## Abstract

What is the shortest path between two data points lying in a high-dimensional space? While the answer is trivial in Euclidean geometry, it becomes significantly more complex when the data lies on a curved manifold—requiring a Riemannian metric to describe the space's local curvature. Estimating such a metric, however, remains a major challenge in high dimensions.

In this work, we propose a method for deriving Riemannian metrics directly from pretrained Energy-Based Models (EBMs)—a class of generative models that assign low energy to high-density regions. These metrics define spatially varying distances, enabling the computation of geodesics—shortest paths that follow the data manifold's intrinsic geometry. We introduce two novel metrics derived from EBMs and show that they produce geodesics that remain closer to the data manifold and exhibit lower curvature distortion, as measured by alignment with ground-truth trajectories. We evaluate our approach on increasingly complex datasets: synthetic datasets with known data density, rotated character images with interpretable geometry, and high-resolution natural images embedded in a pretrained VAE latent space. Our results show that EBM-derived metrics consistently outperform established baselines, especially in high-dimensional settings.

Our work is the first to derive Riemannian metrics from EBMs, enabling data-aware geodesics and unlocking scalable, geometry-driven learning for generative modeling and simulation.

## 1 Introduction

*What is the shortest path between two data points in a high-dimensional space?* In Euclidean geometry, the answer is a straight line. But in modern machine learning, where data often lies on unknown curved manifolds within a high-dimensional space, straight lines slice through regions without data (see linear interp. in Fig. 1). Capturing the true geometry of data is therefore critical in fields where distance-based analyses depend on underlying structure, such as vision [1–3], language [4, 5], biology [6], and cognitive science [7, 8]. Riemannian geometry offers a principled way to navigate these spaces by introducing a smoothly varying local metric, the Riemannian metric, which encodes how space bends and stretches [9]. Within this framework, the shortest path between two points is no longer a straight line, but a geodesic—a curve that follows the intrinsic curvature of the manifold. Computing geodesics requires knowing the underlying Riemannian metric, but estimating such a metric for complex, high-dimensional data remains a major challenge in machine learning.

---

*Equal contribution.

39th Conference on Neural Information Processing Systems (NeurIPS 2025).

A promising strategy for deriving Riemannian metrics is to take a data-driven approach—learning the metric directly from the data itself. This approach estimates the data density and turns it into a Riemannian metric that contracts high-density regions and dilates low-density ones, aligning the geometry with the data manifold [10] (see § 2 for more details). Existing methods, such as kernel-based estimators [11], normalizing flows [12], and density-based constructions [13], have succeeded in low-dimensional settings. However, their performance often degrades in high dimensions, where sparse local sampling makes it hard to capture reliable geometric structure [14, 15]. Meanwhile, recent advances in generative AI [16–18] have produced models capable of capturing complex data distributions in high-dimensional spaces with remarkable accuracy. *If these models can learn the data distribution, can they also reveal its underlying geometry?*

In this article, we answer affirmatively by proposing to derive Riemannian metrics from pretrained Energy-Based Models (EBMs) [16, 19, 20]. EBMs are a flexible class of generative models that define an energy function $E_\theta$, parameterized by a neural network, assigning low energy to likely data points (i.e., $p_\theta(\mathbf{x}) \propto \exp(-E_\theta(\mathbf{x}))$).We show that the energy landscape of an EBM encodes a rich geometric structure and can be leveraged to derive effective Riemannian metrics. Specifically, we introduce two novel conformal Riemannian metrics—metrics that scale the identity by a positive scalar function: $\mathbf{G_{E_\theta}}$ proportional to the energy itself, and $\mathbf{G_{1/p_\theta}}$, proportional to the inverse unnormalized density. We evaluate both against established alternatives ($\mathbf{G_{RBF}}$ [13] and $\mathbf{G_{LAND}}$ [11]) across datasets of increasing complexity—from toy distributions with known geodesics (see § 4.2), to rotated character images where the manifold structure is partially known (see § 4.3), and finally to high-dimensional natural images where no ground truth geometry is available (see § 4.4). Throughout this work, we

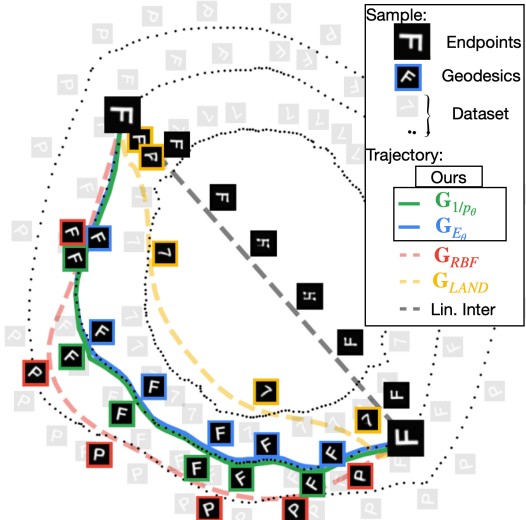

Figure 1: **Geodesics visualization for the URC dataset**. Trajectories and samples are projected in the PCA space for visualization.

.

adopt the common choice of equipping the data space with a density-based Riemannian metric, thereby defining the geometry of the manifold in terms of data concentration. We show that EBM-based metrics yield geodesics that (i) remain closer to the data manifold and (ii) better reflect its intrinsic curvature (see Fig. 1 for a visualization of the geodesics).

Overall, our contributions are summarized as follows:

- We propose a novel framework based on pretrained Energy-Based Models (EBMs) to derive Riemannian metrics. In particular, we introduce two novel conformal metrics $\mathbf{G_{E_\theta}}$ and $\mathbf{G_{1/p_\theta}}$, based on the data log-likelihood and data density, respectively.
- We demonstrate that these EBM-derived metrics yield geodesics that remain closer to the data manifold and better reflect its curvature.
- We show that the proposed EBM-based metrics scale more robustly than prior approaches.

By grounding Riemannian metrics in generative AI, we hope to initiate a new paradigm for understanding and navigating the hidden geometry of high-dimensional data spaces.

## 2   Related Work

**The many facets of data geometry:**   A variety of approaches have been proposed to study the geometry of data:

- *Information Geometry*: This historical approach is rooted in the work of Rao [21] and Amari [22]. It connects statistics and differential geometry by interpreting the Fisher information [23] as a Riemannian metric on the manifold of parameters of a statistical model. In contrast, our

work derives Riemannian metrics directly from the data space using the energy or the likelihood of an EBM.

- *Data-Space induced metrics*: Closer to our work, this approach estimates Riemannian metrics directly from samples. The LAND metric [11] derives a local metric tensor from the empirical covariance of nearby points. The RBF metric [13] defines a conformal metric using an RBF network trained as a parametric KDE, learning centres, widths, and weights so its output forms an unnormalised data density. Both serve as baselines in our study (see § 4.1) and have recently been used for geodesic fitting via flow matching [24]. The (unpublished) work of Perone [25] was also a key inspiration, proposing to build metrics from the score function of a generative model—an idea also explored by Diepeveen et al. [26].
- *Latent-Space induced metrics*: Another line of work uses pullback geometry [27–32], mapping the Euclidean metric from a network's latent space to the data space—typically through the Jacobian of a VAE encoder [33]. While our method operates in the latent space of a VAE in high-dimensional settings, the metric is derived from the energy of the EBM and remains independent of the VAE encoder.
- *Generative modeling on a pre-defined manifold:* Recent approaches such as flow-based models [34, 35] and Schrödinger bridges [36, 37] learn transport paths between distributions, sometimes defined over Riemannian manifolds [38–41]. These methods assume a known, fixed manifold geometry (e.g., a hypersphere) and design generative models to operate within that structure. In contrast, our approach starts from a generative model—an EBM—and derives the Riemannian metric itself from the model, allowing the geometry to emerge from the data.

For a more detailed review of the related work (including topological data analysis, symmetries, computer graphics, or metric learning), see Supp. A and [42, 14].

**Energy-Based Models (EBMs):** EBMs, trained via maximum likelihood [16] (see § 3.2), are particularly well-suited for deriving Riemannian metrics. Their contrastive training, combined with Langevin dynamics sampling, encourages learning a *global* energy landscape that assigns meaningful values across the entire ambient space, including regions far from the data manifold. In contrast, normalizing flows [43] are limited by their invertible architecture [44, 45] and tend to perform poorly on out-of-distribution data [46], sometimes leaking probability mass outside the support [47]. EBMs trained with diffusion losses [48] or distilled from diffusion models [49] generate high-quality samples, but their energy function depends on a time-indexed noise scale, limiting them to local rather than global energy landscapes. This makes them unsuitable for defining a consistent Riemannian metric. Prior work has used the global energy landscape of EBMs trained via maximum likelihood for trajectory planning in robotics [50], though not in the context of geodesics.

## 3 Method

**Notation**: Scalars are denoted by plain lowercase (e.g., x), vectors by bold lowercase (e.g., $\mathbf{x} \in \mathbb{R}^D$), and matrices by bold uppercase (e.g., $\mathbf{X}$). Let $\mathbf{I}$ be the identity matrix of $\mathbb{R}^{D \times D}$. $\mathcal{S}_{++}^D$ is the set of symmetric $D \times D$ positive definite matrices. Let $\mathcal{M}$ be a Riemannian manifold, with tangent space at $\mathbf{x} \in \mathcal{M}$ denoted $\mathcal{T}_{\mathbf{x}}^{\mathcal{M}}$. Herein, we assume that $\mathcal{M}$ is embedded in a $D$-dimensional Euclidian space ($\mathcal{M} \subset \mathbb{R}^D$).

### 3.1 A primer on Riemannian geometry

A Riemannian manifold $(\mathcal{M}, \mathbf{G})$ is a smooth manifold $\mathcal{M}$ (i.e., a set locally homeomorphic to $\mathbb{R}^D$) equipped with a Riemannian metric $\mathbf{G} : \mathcal{M} \to \mathcal{S}_{++}^D$. $\mathbf{G}$ defines a smoothly changing inner product on the tangent space $\mathcal{T}_{\mathbf{x}}^{\mathcal{M}}$ at each point $\mathbf{x} \in \mathcal{M} : \langle \mathbf{u}, \mathbf{v} \rangle_{\mathbf{x}} = \mathbf{u}^{\top} \mathbf{G}(\mathbf{x}) \mathbf{v}$, with $\mathbf{u}, \mathbf{v} \in \mathcal{T}_{\mathbf{x}}^{\mathcal{M}}$ [9]. The length of a curve $\boldsymbol{\gamma} : [0, 1] \to \mathcal{M}$ linking two points $\mathbf{x_0} = \boldsymbol{\gamma}(0)$ and $\mathbf{x_1} = \boldsymbol{\gamma}(1)$ ($\mathbf{x_0}, \mathbf{x_1} \in \mathcal{M}$), is measured as:

$$L(\boldsymbol{\gamma}) = \int_0^1 \sqrt{\langle \dot{\boldsymbol{\gamma}}(t), \dot{\boldsymbol{\gamma}}(t) \rangle_{\boldsymbol{\gamma}(t)}} dt. \tag{1}$$

In Eq. 1, $\dot{\boldsymbol{\gamma}}(t)$ denotes the velocity vector of the curve $\boldsymbol{\gamma}(t)$, which lies in the tangent space at that point (i.e., $\dot{\boldsymbol{\gamma}}(t) \in \mathcal{T}_{\boldsymbol{\gamma}(t)}^{\mathcal{M}}$). The minimizer of Eq. 1 is called a *geodesic*; it represents the (locally) shortest path between $\mathbf{x_0}$ and $\mathbf{x_1}$. In this work, we minimize the kinetic energy functional instead of the length (see Eq. 2). Although both functionals yield the same geodesics up to a parametrization,

minimizing the kinetic energy functional results in a constant Riemannian speed parametrization[2]. This property simplifies optimization and improves numerical stability [9, 13].

$$\gamma^{\star}(t) = \arg\min_{\gamma} \mathcal{E}[\gamma] \ \text{ s.t. } \ \mathcal{E}[\gamma] = \frac{1}{2}\int_0^1 \langle \dot{\gamma}(t), \dot{\gamma}(t)\rangle_{\gamma(t)} \, dt. \tag{2}$$

In the Euclidean case ($\mathcal{M} = \mathbb{R}^D$, $\mathbf{G}(\mathbf{x}) = \mathbf{I}$), $\mathcal{E}$ is equivalent to the kinetic energy of a unit-mass particle moving along $\gamma(t)$, hence the name kinetic energy functional.

To avoid the computational cost of solving Eq. 2 for each new pair $(\mathbf{x}_0, \mathbf{x}_1)$ at inference time, we follow [24] and approximate the geodesic with a neural interpolant $\varphi_\eta$ (with parameters $\eta$).

$$\mathbf{x}_{t,\eta} = (1-t)\mathbf{x}_0 + t\mathbf{x}_1 + 2t(1-t)\varphi_\eta(\mathbf{x}_0, \mathbf{x}_1, t). \tag{3}$$

This parameterization satisfies the boundary conditions ($\mathbf{x}_{0,\eta}{=}\mathbf{x}_0$, $\mathbf{x}_{1,\eta}{=}\mathbf{x}_1$). In Eq. 3, $\varphi_\eta$ serves as a nonlinear correction to the linear path, allowing the learned path to bend toward the data manifold. We train a single interpolant network $\varphi_\eta$ over batches of random endpoint pairs so it can approximate geodesics between arbitrary points (see Algo. 1). Intuitively, our geodesic interpolant begins with a straight line between the endpoints and uses a neural network to compute a smooth curvature relative to this baseline—bending the path toward regions of higher data density, much like pulling a string taut over a curved surface that reflects the geometry of the data. Unlike Kapusniak et al. [24], who use full autodifferentiation to compute $\dot{\mathbf{x}}_{t,\eta}$, we opt for finite difference instead. We found this approach more stable and accurate when using a fine-time discretization.

Although Algo. 1 approximates geodesics for a given metric $\mathbf{G}$, the trajectories may initially deviate from the data manifold—especially early in training, when they are initialized as straight lines in the ambient space. However, if (i) the eigenvalues of $\mathbf{G}$ are large when off-manifold and (ii) small when on-manifold, then the interpolated points $\mathbf{x}_t$ are progressively drawn toward the manifold during optimization [24, 13]. In other words, an effective $\mathbf{G}$ should penalize off-manifold directions and encourage paths through high-density

---

**Algorithm 1:** Training geodesic interpolant

**Input:** Endpoints pairs: ($\{\mathbf{x}_0\}, \{\mathbf{x}_1\}$), Interp. net.: $\varphi_\eta$,
    Metric: $\mathbf{G}$, Time steps: T

dt$=\frac{1}{T-1}$, t=[0:1:dt]

**while** training **do**

 $\mathbf{x}_0 \sim \{\mathbf{x}_0\}$ and $\mathbf{x}_1 \sim \{\mathbf{x}_1\}$  ## sample batch of pairs

 $\mathbf{x}_{t,\eta} = (1-t)\mathbf{x}_0 + t\mathbf{x}_1 + 2t(1-t)\varphi_\eta(\mathbf{x}_0, \mathbf{x}_1, t)$

 $\dot{\mathbf{x}}_{t,\eta} = \dfrac{\mathbf{x}_{t+1,\eta} - \mathbf{x}_{t,\eta}}{dt}$  ## finite difference

 $\mathcal{L}(\eta) = \mathbb{E}_{\mathbf{x}_0,\mathbf{x}_1}\left[\dfrac{1}{2}\sum_{t=0}^{1}\left[\dot{\mathbf{x}}_{t,\eta}^{\top}\mathbf{G}(\mathbf{x}_{t,\eta})\dot{\mathbf{x}}_{t,\eta}\right]dt\right]$

 Update $\eta$ using gradient $\nabla_\eta\mathcal{L}$

---

paths, steering the geodesics along true data geometry. This insight suggests that defining the metric as a decreasing function of the data probability (e.g., $\mathbf{G}(\mathbf{x}) \propto p(\mathbf{x})^{-1}\cdot\mathbf{I}$) can effectively steer trajectories toward high-density regions. In practice, however, the true data distribution is unknown and only observed through samples. In this work, we use an EBM to approximate the data distribution.

## 3.2 Energy-Based Models

Let $p_\mathcal{M}$ be the true data distribution supported on the manifold $\mathcal{M}$, such that $\int_{\mathbf{x}\in\mathcal{M}} p_\mathcal{M}(\mathbf{x})d\mathbf{x} = 1$. In practice, we do not have access to $p_\mathcal{M}$ directly, but only to a finite set of samples $\mathcal{D} = \{\mathbf{x}_i\}_{i=1}^N$ drawn from it. These samples define the empirical distribution $p_\mathcal{D}$, which we use to train our models.

Energy-Based Models (EBMs) provide a flexible framework for modeling complex, unnormalized probability distributions—making them particularly well-suited for data concentrated on low-dimensional manifolds. Here we define the energy function $E_\theta(\mathbf{x}) \in \mathbb{R}$, parameterized with a neural network with weights $\theta$. This energy induces a probability distribution of the form:

$$p_\theta(\mathbf{x}) = \frac{\exp\big(-E_\theta(\mathbf{x})\big)}{Z(\theta)} \ \text{ where } \ Z(\theta) = \int \exp\big(-E_\theta(\mathbf{x})\big)d\mathbf{x}. \tag{4}$$

---

[2]With length fixed, the strictly convex energy $E = \frac{1}{2}\int_0^1 v(t)^2 dt$ attains its minimum—by Jensen's inequality—only when the speed $v(t)$ is constant.

Our goal is to train the EBM so that $p_\theta$ approximates the data distribution $p_\mathcal{M}$. To do so, we minimize the negative log-likelihood w.r.t to the empirical distribution: $\mathcal{L}_{ML}(\theta) = \mathbb{E}_{\mathbf{x} \sim p_D}[-\log p_\theta(\mathbf{x})]$. Although the partition function $Z(\theta)$ is intractable, previous works have shown that the gradient of this objective can be estimated without computing $Z(\theta)$ explicitly [51, 52] (see Supp. B.1 for the demonstration), a loss known as *contrastive divergence*:

$$\nabla_\theta \mathcal{L}_{ML} \approx \mathbb{E}_{\mathbf{x}^+ \sim p_\mathcal{D}}[E_\theta(\mathbf{x}^+)] - \mathbb{E}_{\mathbf{x}^- \sim p_\theta}[E_\theta(\mathbf{x}^-)] \tag{5}$$

where $\mathbf{x}^+$ are data samples and $\mathbf{x}^-$ are samples drawn from the model distribution $p_\theta$ using Langevin dynamics. We adopt the training procedure of [16], which is known to scale well (see Supp. B for the full pseudo-code). From this point on, we refer to $E_\theta$ as a pre-trained energy function.

EBM can be hard to train in high-dimensional pixel space, especially because of the sampling procedure [53–55]. For complex tasks, we follow standard practice and operate in the latent space of a pretrained VAE [56], where all baselines are evaluated for fairness. To improve the EBM training training stability, we regularize the contrastive divergence loss with a denoising term, which preserves the global structure of the energy landscape while enhancing convergence—a technique we find both effective and broadly applicable.

### 3.3 EBM-derived Riemannian Metrics

Here, we describe the EBM-derived metrics $\mathbf{G}_{\mathbf{E}_\theta}$, $\mathbf{G}_{\mathbf{1}/\mathbf{p}_\theta}$. For details on the baseline Riemannian metrics $\mathbf{G}_{\mathbf{LAND}}$, $\mathbf{G}_{\mathbf{RBF}}$, see § 4.1. To ensure a fair comparison —and following standard practice in the field [42, 57, 58]— all metrics are cast using a shared parametric form:

$$\mathbf{G}(\boldsymbol{x}) = \begin{cases} \alpha \, \mathbf{h}(\boldsymbol{x}) + \beta & \text{for } \mathbf{G}_{\mathbf{E}_\theta}, \\ (\alpha \, \mathbf{h}(\boldsymbol{x}) + \beta)^{-1} & \text{for } \mathbf{G}_{\mathbf{1}/\mathbf{p}_\theta}, \mathbf{G}_{\mathbf{LAND}}, \mathbf{G}_{\mathbf{RBF}}, \end{cases}$$

where $\mathbf{h}(\mathbf{x})$ is a metric-specific, positive-definite function (either scalar, diagonal, or matrix), and $\alpha, \beta$ are calibration constants. These constants are chosen so that the metric scale to $\mathbf{I}$ on the data manifold and to $10^3 \cdot \mathbf{I}$ in low-density regions[3]. This allows fair comparison across metric choices without introducing significant sensitivity to hyperparameter tuning. Further details about the metric calibration procedure are provided in Supp. C.1. Importantly, all EBM-derived metrics are *conformal*, they take the form $\lambda(\mathbf{x})\mathbf{I}$, where $\lambda$ is a scalar function. In other words, they scale the identity matrix uniformly in all directions, resulting in isotropic metrics:

- $\mathbf{G}_{\mathbf{E}_\theta}$ defines a Riemannian metric by directly scaling the raw energy of a pretrained EBM. This is the simplest —yet surprisingly effective—formulation we consider:

$$\mathbf{G}_{\mathbf{E}_\theta}(\mathbf{x}) = (\alpha * E_\theta(\mathbf{x}) + \beta) \cdot \mathbf{I}. \tag{6}$$

  Intuitively, high-energy (low-density) regions receive a larger metric, penalizing movement away from the data. Note that $E_\theta$ is an affine rescaling of the negative log-likelihood $-\log p_\mathcal{D}$.
- $\mathbf{G}_{\mathbf{1}/\mathbf{p}_\theta}$ leverages the inverse of an unnormalized probability estimate:

$$\mathbf{G}_{\mathbf{1}/p_\theta}(\mathbf{x}) = (\alpha * \exp(-E_\theta(\mathbf{x})) + \beta)^{-1} \cdot \mathbf{I}. \tag{7}$$

  Compared to $\mathbf{G}_{\mathbf{E}_\theta}$, this metric applies an inverse to a decreasing exponential, forming a strong barrier against low-density regions. It stays small near the data manifold but rises sharply elsewhere, acting as a repulsive force. Its key advantages are: (i) a clear probabilistic interpretation via the unnormalized density, and (ii) direct comparability to $\mathbf{G}_{\mathbf{LAND}}$ and $\mathbf{G}_{\mathbf{RBF}}$ as they share the same inverse form.

In the next section, we introduce the baseline Riemannian metrics used for comparison. We also empirically evaluate their behavior across datasets of increasing complexity, focusing on how they capture the underlying manifold and shape geodesic paths.

## 4 Experiments

### 4.1 Baseline Riemannian Metrics

$\mathbf{G}_{\mathbf{RBF}}$ [13, 24] and $\mathbf{G}_{\mathbf{LAND}}$ [11] are established metrics from the Riemannian geometry literature:

---

[3]Note that this multiplicative factor amounts to a change of unit, to ensure reasonable scaling of the lengths, but the induced geodesics are only determined by the ratio $\alpha/\beta$.

- $G_{\text{LAND}}$, also known as the LAND metric [11], is a nonparametric Riemannian metric that adapts to the local geometry of the dataset. Around each point $\mathbf{x}$, it estimates a Gaussian distribution by weighting all data points $\{\mathbf{x}_i\}_{i=1}^N$ according to their distance to $\mathbf{x}$:

$$G_{\text{LAND}}(\mathbf{x}) = (\alpha \, \text{diag}(\mathbf{h}(\mathbf{x})) + \beta \mathbf{I})^{-1} \text{ s.t } h^{(j)}(\mathbf{x}) = \sum_{i=1}^N (x_i^{(j)} - x^{(j)})^2 \exp\left(-\frac{\|\mathbf{x} - \mathbf{x_i}\|^2}{2\sigma^2}\right) \quad (8)$$

  Here, $h^{(j)}(\mathbf{x})$ measures the local variance along dimension $j$, weighted by a Gaussian kernel with bandwidth $\sigma$. $G_{\text{LAND}}$ is the only diagonal (i.e., non-conformal) metric we consider, allowing it to model local anisotropy. While flexible and model-free, LAND has practical drawbacks: it requires the full dataset at inference, is sensitive to the choice of $\sigma$, and can behave non-smoothly near sharp neighborhood transitions (see Supp. C.2 for examples).

- $G_{\text{RBF}}$ is a conformal Riemannian metric in which $h$ is a weighted sum of Radial Basis Functions (RBFs) centered on $K$ cluster centroids $\{\hat{\mathbf{x}}_k\}_{k=1}^K$ computed via K-means [13]:

$$G_{\text{RBF}}(\mathbf{x}) = (\alpha \cdot h(\mathbf{x}) + \beta)^{-1} \cdot \mathbf{I}, \quad h(\mathbf{x}) = \sum_{k=1}^K w_k \exp\left(-\frac{1}{2} \cdot \lambda_k \|\mathbf{x} - \hat{\mathbf{x}}_k\|^2\right).$$

  The weights $w_k$ are trained so that $h(\mathbf{x}) \approx 1$ on the data manifold, and $\lambda_k$ is set from inter-cluster distances (see Supp. C.3). This yields a smooth, efficient approximation of the data geometry and scales better than LAND [24]. However, it may miss fine-grained structure, especially in regions of complex or uneven density. Like other methods based on Euclidean distance (and K-means), it suffers from the curse of dimensionality. Its accuracy depends on $K$, $\lambda_k$, and centroid placement (illustrated in Supp. C.3).

The scaling constants $(\alpha, \beta)$ are introduced to ensure consistent dynamic range across metrics and have minimal impact on convergence or geodesic quality; the number of discretization steps ($T = 100$) is chosen as a trade-off between efficiency and accuracy, consistent with prior work. We evaluate $G_{1/\mathbf{p}_\theta}$, $G_{\mathbf{E}_\theta}$, $G_{\text{RBF}}$, and $G_{\text{LAND}}$ on three datasets of increasing complexity. Circular Mixture of Gaussians offers full control and ground-truth geodesics. The rotated characters dataset is higher-dimensional but still allows quantitative evaluation. Animal Faces is made of higher-dimensional images but with no ground truth. This progression tests metric performance as data complexity grows. The code to reproduce all our experiments is available at `https://github.com/VictorBoutin/RiemannEBM`.

### 4.2 Circular Mixture of Gaussians

We consider two toy datasets built using a mixture of Gaussians arranged along a semicircle. In the first, called Uniform Circular Gaussians (UCG), the Gaussian components have equal weights (see Fig. 2a). In the second, Weighted Circular Gaussians (WCG), the weights are non-uniform, with higher density near the center of the arc, as reflected by the contour intensity shown in Fig. 2c. For both datasets, we have access to the closed-form probability distribution of the data, denoted $p_\mathcal{M}$ (see Supp. D.1 for details of $p_\mathcal{M}$). We first train an Energy-Based Model (EBM) on each dataset to derive the metrics $G_{\mathbf{E}_\theta}$ and $G_{1/\mathbf{p}_\theta}$ (see Supp. D.2 for training details). Then, we apply Algo. 1 to both datasets using all Riemannian metrics described above. Additionally, we include two baseline Riemannian metrics derived directly from the true distribution $p_\mathcal{M}$:

$$\mathbf{G}_{\text{E}_\mathcal{M}}(\mathbf{x}) = -\alpha * \log p_\mathcal{M}(\mathbf{x})\mathbf{I} + \beta \quad \text{and} \quad \mathbf{G}_{1/p_\mathcal{M}}(\mathbf{x}) = (\alpha * p_\mathcal{M}(\mathbf{x}) + \beta)^{-1} \cdot \mathbf{I} \quad (9)$$

Eq. 9 uses calibration constants $\alpha$ and $\beta$, computed as in other metrics. Some geodesics obtained for the 6 different metrics are shown in Fig. 2a and Fig. 2c for the UCG and WCG datasets, respectively. We refer the reader to Supp. D for details on network architectures and hyperparameters.

To evaluate geodesic quality, we use two evaluation metrics. The first is the accumulated probability along the geodesic path, $p_\mathcal{M}(\gamma^\star) = \sum_{t=1}^T p_\mathcal{M}(\mathbf{x}_{t,\eta^\star})$. It measures how closely the trajectory aligns with the data manifold — the higher the better. The second is the RMSE to a baseline geodesic computed using the true distribution $p_\mathcal{M}$, matched by metric type (e.g., $\mathbf{G}_{\text{E}_\theta}$ vs. $\mathbf{G}_{\text{E}_\mathcal{M}}$). All quantitative results are averaged over $1,000$ geodesics with distinct endpoints (See Fig. 2b and d). $G_{\mathbf{E}_\theta}$ achieves the highest accumulated probability, indicating closest alignment with the data manifold, while $G_{1/\mathbf{p}_\theta}$ yields the lowest RMSE to its baseline—best approximating the ground-truth geodesic. Both EBM-based metrics consistently outperform other methods across evaluation criteria.

To test how different metrics behave when the density varies along the data manifold, we switch

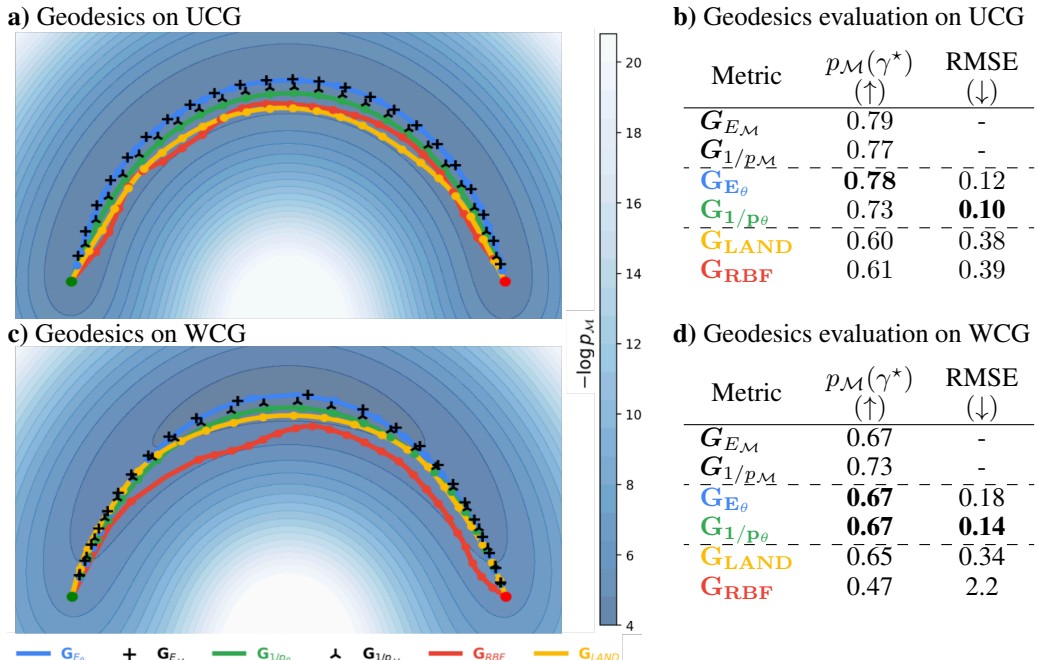

**a)** Geodesics on UCG

**b)** Geodesics evaluation on UCG

| Metric | $p_{\mathcal{M}}(\gamma^\star)$ ($\uparrow$) | RMSE ($\downarrow$) |
|---|---|---|
| $\boldsymbol{G}_{E_{\mathcal{M}}}$ | 0.79 | - |
| $\boldsymbol{G}_{1/p_{\mathcal{M}}}$ | 0.77 | - |
| $\mathbf{G}_{\mathbf{E}_\theta}$ | **0.78** | 0.12 |
| $\mathbf{G}_{\mathbf{1/p}_\theta}$ | 0.73 | **0.10** |
| $\mathbf{G}_{\mathbf{LAND}}$ | 0.60 | 0.38 |
| $\mathbf{G}_{\mathbf{RBF}}$ | 0.61 | 0.39 |

**c)** Geodesics on WCG

**d)** Geodesics evaluation on WCG

| Metric | $p_{\mathcal{M}}(\gamma^\star)$ ($\uparrow$) | RMSE ($\downarrow$) |
|---|---|---|
| $\boldsymbol{G}_{E_{\mathcal{M}}}$ | 0.67 | - |
| $\boldsymbol{G}_{1/p_{\mathcal{M}}}$ | 0.73 | - |
| $\mathbf{G}_{\mathbf{E}_\theta}$ | **0.67** | 0.18 |
| $\mathbf{G}_{\mathbf{1/p}_\theta}$ | **0.67** | **0.14** |
| $\mathbf{G}_{\mathbf{LAND}}$ | 0.65 | 0.34 |
| $\mathbf{G}_{\mathbf{RBF}}$ | 0.47 | 2.2 |

Legend: $\mathbf{G}_{E_\theta}$ — $\boldsymbol{G}_{E_{\mathcal{M}}}$ (+) — $\mathbf{G}_{1/p_\theta}$ — $\boldsymbol{G}_{1/p_{\mathcal{M}}}$ — $\mathbf{G}_{RBF}$ — $\mathbf{G}_{LAND}$

Figure 2: **Geodesics on UCG and WCG datasets. (a, c)**: Some geodesics obtained on UCG **(a)** and WCG **(c)**, for 6 different Riemannian metrics. The contour plots represent the energy landscape given by $-\log p_{\mathcal{M}}$. **(b, d)** Quantitative evaluation of geodesics on UCG **(b)** and WCG **(d)**. We report (i) the accumulated probability along the geodesic (the higher the better) and ii) RMSE between each geodesic and its corresponding baseline (i.e., $\boldsymbol{G}_{E_{\mathcal{M}}}$ for $\mathbf{G}_{\mathbf{E}_\theta}$, and $\boldsymbol{G}_{1/p_{\mathcal{M}}}$ for $\mathbf{G}_{\mathbf{1/p}_\theta}$, $\mathbf{G}_{\mathbf{LAND}}$ and $\mathbf{G}_{\mathbf{RBF}}$). See Supp. D.3 for the 2-$\sigma$ error.

from the uniformly populated UCG semicircle to the Weighted Circular Gaussian (WCG), whose samples cluster near the arc's centre. As shown in Fig. 3, log-based metrics ($\mathbf{G}_{\mathbf{E}_\theta}$, $\boldsymbol{G}_{E_{\mathcal{M}}}$) accentuate the manifold curvature more than $1/p$-based ones ($\mathbf{G}_{\mathbf{1/p}_\theta}$, $\mathbf{G}_{\mathbf{RBF}}$, $\mathbf{G}_{\mathbf{LAND}}$, $\boldsymbol{G}_{1/p_{\mathcal{M}}}$), producing larger steps in high-density regions. This is because $-\log p$ diverges as $p \to 0$, amplifying distortions and speed variations.

### 4.3 Rotated Characters

We use an image dataset of seven rotated, non-symmetric characters in two variants: Uniform Rotated Characters (URC), with evenly distributed angles, and Biased Rotated Characters (BRC), concentrated near $0°$. In this subsection, all computations are done in the $64$-dimensional latent space of a regularized autoencoder trained with a triplet loss, ensuring that small angular differences yield short latent distances. This setup provides a unique middle ground: although the underlying Riemannian metric is

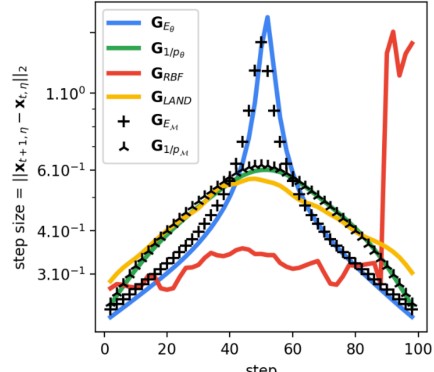

Figure 3: **Step size along geodesics in the WCG dataset**. Log-based metrics ($\mathbf{G}_{\mathbf{E}_\theta}$ and $\boldsymbol{G}_{E_{\mathcal{M}}}$) produce sharper variations, reflecting stronger sensitivity to density curvature.

unknown, we can treat the smooth in-plane rotation between two instances of the same character as a proxy for the ground-truth geodesic. Thanks to the triplet loss, the latent space is structured so that nearby points correspond to slight rotations of the same character, making the shortest path between two orientations a meaningful approximation of the true geodesic in the task-relevant transformation space. Separate EBMs and interpolant networks are trained for each dataset variant. Full experimental details (datasets, architectures, and hyperparameters) are provided in Supp. E.

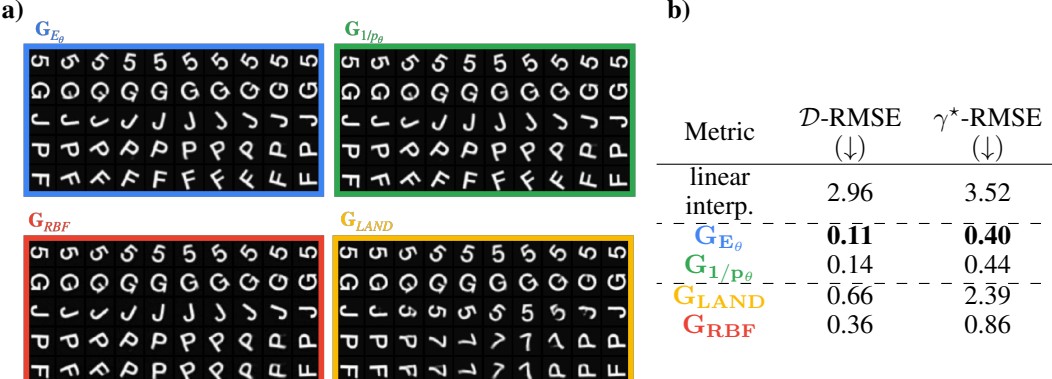

a)

$\mathbf{G}_{E_\theta}$   $\mathbf{G}_{1/p_\theta}$

$\mathbf{G}_{RBF}$   $\mathbf{G}_{LAND}$

b)

| Metric | $\mathcal{D}$-RMSE ($\downarrow$) | $\gamma^\star$-RMSE ($\downarrow$) |
|---|---|---|
| linear interp. | 2.96 | 3.52 |
| $\mathbf{G}_{\mathbf{E}_\theta}$ | **0.11** | **0.40** |
| $\mathbf{G}_{\mathbf{1}/\mathbf{p}_\theta}$ | 0.14 | 0.44 |
| $\mathbf{G}_{\mathbf{LAND}}$ | 0.66 | 2.39 |
| $\mathbf{G}_{\mathbf{RBF}}$ | 0.36 | 0.86 |

Figure 4: **Geodesics on the URC dataset. (a)** Geodesics computed with different Riemannian metrics, projected into pixel space for visualization. $\mathbf{G}_{\mathbf{RBF}}$ and $\mathbf{G}_{\mathbf{LAND}}$ often deviate from the intended path, sometimes drifting toward other characters (e.g., the letter $F$). **(b)** Quantitative evaluation using two metrics: (i) $\mathcal{D}$-RMSE, which measures proximity to the dataset manifold, and (ii) $\gamma$-RMSE, which measures the deviation from an ideal smooth rotation. See Supp. E.6 for the 2-$\sigma$ error.

In Fig. 4a, we visualize geodesics computed on the URC dataset, projected back into pixel space (see Supp. E.5 for additional results on both URC and BRC). EBM-based metrics ($\mathbf{G}_{\mathbf{E}_\theta}$ and $\mathbf{G}_{\mathbf{1}/\mathbf{p}_\theta}$) yield smooth rotations that preserve character identity, while $\mathbf{G}_{\mathbf{RBF}}$ and especially $\mathbf{G}_{\mathbf{LAND}}$ often deviate from the intended trajectory. To illustrate these failures, Fig. 1 shows all geodesics projected into PCA space for a case involving the letter F. While $\mathbf{G}_{\mathbf{E}_\theta}$ and $\mathbf{G}_{\mathbf{1}/\mathbf{p}_\theta}$ remain on the manifold of rotated F instances, linear interpolation cuts through low-density regions, and $\mathbf{G}_{\mathbf{RBF}}$ and $\mathbf{G}_{\mathbf{LAND}}$ drift toward other character classes. To quantify this, we use two metrics: $\mathcal{D}$-RMSE, which measures the average distance from each geodesic point to its nearest neighbor in the dataset—lower values indicate better adherence to the data manifold; and $\gamma$-RMSE, which evaluates how closely the geodesic follows an ideal smooth rotation between endpoints. All results are averaged over $1,000$ geodesics with random endpoint orientations. As shown in Fig. 4b, EBM-based metrics consistently outperform others; $\mathbf{G}_{\mathbf{RBF}}$ performs reasonably well, while $\mathbf{G}_{\mathbf{LAND}}$

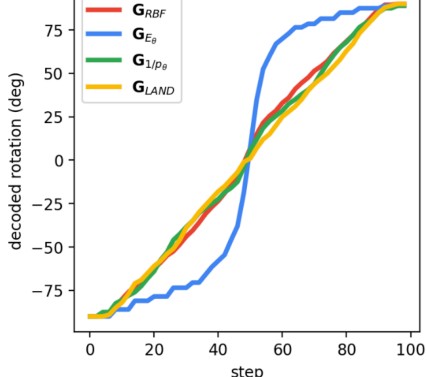

Figure 5: **Step size along geodesics in the WCG dataset**. Log-based metric ($\mathbf{G}_{\mathbf{E}_\theta}$) produces sharper variations, reflecting stronger sensitivity to density curvature.

shows large deviations on both metrics. Overall, these results suggest that EBM-based metrics scale more effectively to high-dimensional data than alternative approaches.

As in the previous section, we examine how different metrics influence a geodesic's ability to follow the manifold's curvature. We focus on the BRC dataset, where orientations are biased toward $0°$, creating sharper curvature near that region. To assess this, we decode the orientation at each time step along geodesics connecting fixed endpoints. As shown in Fig. 5, geodesics under $\mathbf{G}_{\mathbf{E}_\theta}$ rotate significantly faster near $0°$ than those under $\mathbf{G}_{\mathbf{1}/\mathbf{p}_\theta}$ and $\mathbf{G}_{\mathbf{RBF}}$, reflecting stronger sensitivity to density variations.

At first glance, it may seem counterintuitive that trajectories following the geodesics move faster in high-density regions. However, this is consistent with minimizing the kinetic energy $\mathcal{E}$ in Eq.2, which enforces constant Riemannian speed (i.e., the quantity $||\dot{\gamma}(t)||_{\gamma(t)}$ is preserved along the trajectory) but not a constant Euclidean speed (i.e., $||\dot{\gamma}(t)||$ is not constant). Since EBM-derived metrics assign lower Riemannian cost in high-density regions, maintaining constant Riemannian speed requires moving faster in Euclidean terms through these regions. The faster rotation near $0°$, observed in Fig.5 and Fig. 3, thus reflects the lower Riemannian cost of traveling through high-density regions. These results confirm and extend our previous findings: metrics based on energy (i.e., proportional to $-\log p$) more effectively capture the curvature of the data manifold.

## 4.4 Animal Faces

We now evaluate our method on the Animal Faces High Quality (AFHQ) dataset [59], using the latent space of the pretrained Stable Diffusion v1 VAE [18] (latent dimension: $4 \times 16 \times 16$). An EBM is trained to model the distribution of latent codes, and Algo. 1 is used to compute geodesics between a cat and a dog representation. We compare the resulting paths to two baselines: (i) linear interpolation and (ii) spherical interpolation (slerp) [60], which is known to better preserve the structure of VAE latent spaces under Gaussian priors (see Supp. F.6). Full experimental details are in Supp. F.

Fig. 6 illustrates geodesics computed in the latent space of a pretrained VAE and projected back into image space (see F.5 for additional samples as well as samples for $\mathbf{G_{LAND}}$ and linear interpolation). Qualitatively, we observe that geodesics computed with the $\mathbf{G_{1/p_\theta}}$ metric best adhere to the data manifold. The $\mathbf{G_{E_\theta}}$ metric also shows noticeable improvements over the other metrics. Despite extensive tuning, $\mathbf{G_{RBF}}$ and $\mathbf{G_{LAND}}$ produce trajectories only slightly better than linear interpolation—suggesting these parametric metrics struggle to scale in high dimensions, consistent with prior findings [11, 13].

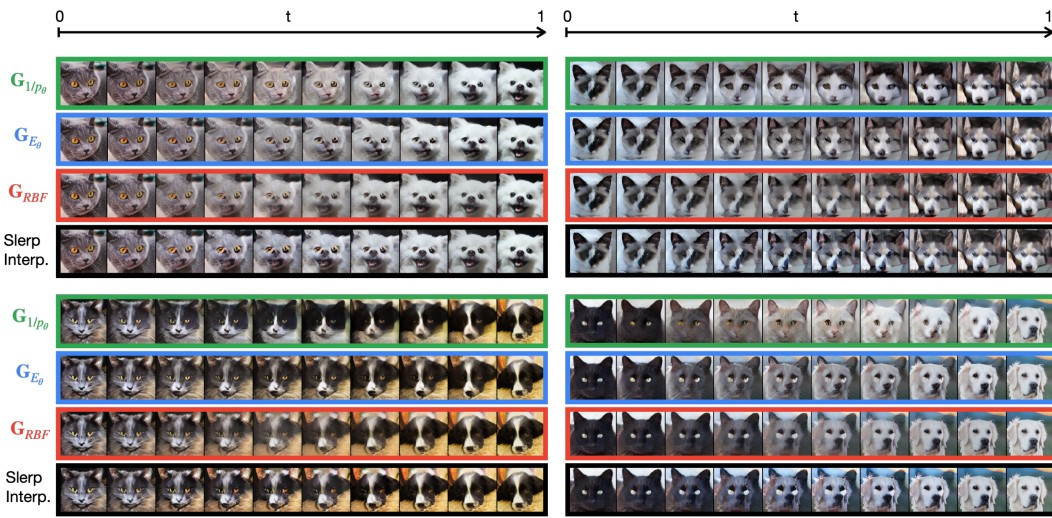

Figure 6: **Geodesics on the AFHQ dataset.** Each block shows an interpolated trajectory between two animal images (cats and dogs), projected back into image space for visualization. We compare geodesics computed with two EBM-based metrics ($\mathbf{G_{1/p_\theta}}$, $\mathbf{G_{E_\theta}}$), a parametric RBF-based metric ($\mathbf{G_{RBF}}$), and spherical interpolation (slerp). Results using $\mathbf{G_{LAND}}$, linear interpolation, and additional examples are provided in Supp. F.5.

To quantitatively assess geodesic quality, we report FID scores [61] in Table 1, computed over $50,000$ trajectories that interpolate from randomly chosen cat images to randomly chosen dog images. The results are consistent with qualitative observations: $\mathbf{G_{1/p_\theta}}$ and $\mathbf{G_{E_\theta}}$ yield the lowest FIDs, followed by the model-free slerp baseline, then $\mathbf{G_{RBF}}$, $\mathbf{G_{LAND}}$, and linear interpolation. Note that the FID measures how aligned individual samples are with the training distribution—on-manifold alignment—but does not assess whether the full trajectory respects the true manifold curvature. Unfortunately, AFHQ lacks ground-truth geometry for such evaluation.

| Metric | FID ($\downarrow$) |
|---|---|
| Linear interp. | 42.47 |
| Slerp interp. | 32.67 |
| $\mathbf{G_{E_\theta}}$ | 20.79 |
| $\mathbf{G_{1/p_\theta}}$ | **16.47** |
| $\mathbf{G_{LAND}}$ | 39.17 |
| $\mathbf{G_{RBF}}$ | 37.98 |

Table 1: **FID along geodesics for different Riemannian metrics**. FID is computed at each trajectory point to assess on-manifold alignment. See Supp. F.4 for the 2-$\sigma$ error.

## 5 Conclusion

In this work, we use pretrained Energy-Based Models (EBMs) to derive conformal Riemannian metrics, $\mathbf{G_{E_\theta}}$ and $\mathbf{G_{1/p_\theta}}$, and we compare them to established alternatives ($\mathbf{G_{LAND}}$ [11] and

$\mathbf{G_{RBF}}$ [13]). On both synthetic and high-dimensional data, EBM-derived metrics yield geodesics that stay closer to the data manifold and better capture its curvature—especially with $\mathbf{G_{E_\theta}}$.

We focus on conformal metrics, which scale the identity by a scalar field to encode density. While more complex, non-conformal and anisotropic metrics (e.g., the Stein metric [25]) are accessible from the EBM score, we found that conformal metrics offer comparable performance with simpler interpretation and reduced computational cost, justifying our focus in this work. Future work may explore these extensions with regularization or structural priors to ensure smoothness and scalability (See Supp.G for a discussion of limitations and Supp.H for broader impact). To keep computational cost manageable, we train the EBM in the latent space of a pretrained autoencoder and compute geodesics using finite-difference optimization, two design choices that substantially reduce complexity and memory use without compromising performance.

Although this article is primarily methodological, it points to promising applications. One example is the mental rotation task, in which humans mentally rotate objects to match a target [62]. In such experiments, reaction times tend to decrease with training [63], suggesting that repeated exposure sharpens internal representations around training examples. These refined representations may concentrate in high-density regions, where mental transformations occur more quickly. As shown in Fig. 3 and 5, our geodesics naturally accelerate in such high-density regions, echoing these psychophysical findings. Modeling mental simulation as geodesics on Riemannian manifolds shaped by a generative model offers a principled computational framework to understand human cognition. It provides a way to formalize and test the hypothesis that the human cognition relies on generative models to support flexible inference [64–68]. Our approach is also particularly relevant for neuroscience, where datasets are high-dimensional, often sparsely sampled, and where understanding the geometry of neural population activity is central to scientific insight. In such settings, high-fidelity geodesics are essential for capturing the true structure of neural trajectories—approximations may distort the manifold and lead to misinterpretation of brain dynamics. While training EBMs is costly, the benefits in terms of interpretability and geometric accuracy make this approach compelling for applications where precision is critical.

As machine learning models are increasingly used to capture complex data distributions, understanding the geometry of their latent spaces becomes essential. Our work contributes to this effort by showing that geometry can serve as a useful tool for building models that better reflect data structure, align with human perception, and shed light on cognitive processes.

## Acknowledgments

This work was supported by ANR-3IA Artificial and Natural Intelligence Toulouse Institute (ANR-19-PI3A-0004). Part of this work was carried out within the DEEL project, which is part of IRT Saint Exupéry and the ANITI AI cluster. The authors acknowledge the financial support from DEEL's industrial and academic members and the "France 2030" program (NR-10-AIRT-01 and ANR-23-IACL-0002). Additional support for TS provided by ONR (N00014-24-1-2026 and REPRISM MURI N00014-24-1-2603) and NSF (IIS-2402875).

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

# A Extended related work

Several tools have been developed to study the geometrical properties of distributions. We survey some prominent approaches below.

**Information geometry**, initiated by the seminal works of [21, 22], was the first to apply ideas from differential geometry to the field of statistics. Unlike our present work, the goal was not to understand the geometry of the data $x$, but rather to understand the geometry of a smooth manifold $\theta \in \Theta$ of parameters of an estimator $p_\theta$. In particular, starting from the Taylor expansion of *reverse* Kullback-Leibler [69] divergence to $p_\theta$, in the neighborhood of $p_\theta$ itself, with $\theta' = \theta + \epsilon$, we get

$$D_{KL}(p_{\theta'}\|p_\theta) \approx \underbrace{D_{KL}(p_\theta\|p_\theta)}_{=0} + \underbrace{\nabla_1 D_{KL}(p_\theta\|p_\theta)}_{=\mathbf{0}} \epsilon + \epsilon^T \nabla_1^2 D_{KL}(p_\theta\|p_\theta)\epsilon. \qquad (10)$$

One can show that, since $\theta' = \theta$ is the global minimum of this function, the first-order term vanishes and the second-order term $\nabla_1^2 D_{KL}(p_\theta\|p_\theta)$ must be a positive definite form - i.e an inner product. This quantity, called *Fisher information* [23], gives $\Theta$ the structure of a Riemannian manifold. The Riemannian gradient associated with this manifold yields a second-order optimization method coined *natural gradient descent* [70], that has been proved helpful in deep learning [71]. Our method inherits some spirit of this approach, since we define a local inner product as a function of the density, to give a Riemannian structure to the data manifold. However, we focus on the geometry of the *data $x$*, not the geometry of the model's parameters $\boldsymbol{\theta}$.

**Riemannian structure of data manifolds** has already been proposed in the past. For example, the seminal LAND metric [11] is a non-conformal metric built from the samples, with the intent of generalizing multivariate normal distributions to manifolds. The RBF metric [13] is a conformal metric, derived from a kernel density estimator, with some learnable coefficients. More recently, Kapusniak et al. [24] proposed to use those metrics and learn a flow matching algorithm to fit geodesics in the data manifold. The Jacobian of a generative model also defines a metric [33]. The (unpublished) work of Perone [25] has been inspirational for our contribution. They use the Stein score function to build the metric, an approach also chosen by [26] - although restricted to unimodal densities.

**Pullback geometry of latent manifolds** is an active research area. [72] studies the manifold of representations of a given network, while [30] builds a generative autoencoder to represent the manifold. Shortest paths are computed with fixed-point methods [31], or using a discrete graph [32]. While we may rely on the latent space of a VAE for some challenging tasks, studying latent representations of a neural network is beyond the scope of our work.

**On-manifold generative models** can be found in the literature. For example, we can mention flow and bridge matching approaches [34, 35], which learn a flow between a source and a target distribution, including on Riemannian manifolds [38, 39]. In particular, the Schrödinger bridge [36, 37] focuses on an optimization problem involving paths in the space of probability distributions, and was also generalized to non-Euclidean geometries [40, 41]. These works differ significantly from ours: they assume the Riemannian manifold to be given, not chosen, and they build a generative model on top of it. To the contrary, given a special class of generative models to represent the data, we *choose* the metric to build the manifold.

**Topological data analysis** [73, 74] studies the topological properties of the data manifold. This field aims to estimate some topological invariants such as the Euler characteristic [75] and persistent Betti numbers [76] (which are the number of connected components, number of closed loops, etc.) from a finite sample. It relies on tools such as persistent homology [77–79] to design algorithms. This approach typically focuses on the *global* properties: it assumes that the data accumulate on a well-defined manifold, from which these high-level features must be computed. To the contrary, our approach focuses on the *local* structure defined by the metric, while the global structure is inherited from the induced geodesics. Furthermore, we consider the whole ambient space for our manifold, tweaking only the metric to account for low-density regions.

**Symmetries and geometry in representations** have gathered considerable attention from the deep learning community, warranting no fewer than 3 workshops at Neurips alone [4]. Symmetries are

---

[4]`https://www.neurreps.org`

operations under which a structure is left invariant, or equivariant. In particular, some neural architectures are leveraged to reflect priors about the underlying symmetries of the data [80–84]. In other cases, symmetries are discovered and learned from observations [85–87]. Unlike these approaches, we do not seek symmetries in data, and we make minimal assumptions about the model; we are mainly interested in the density to build the structure.

**Non-Euclidean 2D and 3D manifolds** are first-class citizens in computer graphics. The works of [88, 89] define a way to find shortest paths over such manifolds. However, this requires solving the Eikonal equation, which is prohibitively expensive in high dimensions or restricted to Euclidean geometries [90]. Geodesics can be learned, but this is restricted to low dimensions [89]. These setups are beyond the scope of our work, as we focus on higher-dimensional and sparsely populated spaces, and no discrete meshes can be built from samples.

**Metric learning** (or *distance learning*) is another field whose purpose is to learn a distance function between samples, typically in a weakly-supervised manner with contrastive losses [91–93]. Often, these distances cannot be realized as a geodesic distance and are intended for a specific task, like classification or retrieval.

# B  Energy-Based Model

## B.1  Derivation of the Gradient of the EBM Log-Likelihood

The demonstration below is adapted from [52] to fit our notation. Even though this mathematical derivation is not crucial for a good understanding of our work, we include it to make sure our article is self-contained and complete.

We consider an Energy-Based Model (EBM) defining a probability distribution via the Boltzmann form:

$$p_\theta(\mathbf{x}) = \frac{\exp(-E_\theta(\mathbf{x}))}{Z(\theta)} \quad \text{with} \quad Z(\theta) = \int \exp(-E_\theta(\mathbf{x})) \, d\mathbf{x}.$$

Our goal is to minimize the negative log-likelihood with respect to the empirical data distribution $p_\mathcal{D}$:

$$\mathcal{L}_{\text{ML}}(\theta) = \mathbb{E}_{\mathbf{x} \sim p_\mathcal{D}}[-\log p_\theta(\mathbf{x})].$$

We first expand the log-probability:

$$-\log p_\theta(\mathbf{x}) = E_\theta(\mathbf{x}) + \log Z(\theta).$$

Taking the gradient with respect to $\theta$:

$$\nabla_\theta \mathcal{L}_{\text{ML}} = \mathbb{E}_{\mathbf{x} \sim p_\mathcal{D}} \left[ \nabla_\theta E_\theta(\mathbf{x}) + \nabla_\theta \log Z(\theta) \right].$$

The derivative of the log-partition function could be simplified:

$$\begin{aligned}
\nabla_\theta \log Z(\theta) &= \frac{1}{Z(\theta)} \nabla_\theta Z(\theta) \\
&= \frac{1}{Z(\theta)} \nabla_\theta \int \exp(-E_\theta(\mathbf{x})) \, d\mathbf{x} \\
&= -\frac{1}{Z(\theta)} \int \exp(-E_\theta(\mathbf{x})) \nabla_\theta E_\theta(\mathbf{x}) \, d\mathbf{x} \\
&= -\int p_\theta(\mathbf{x}) \nabla_\theta E_\theta(\mathbf{x}) \, d\mathbf{x} \\
&= -\mathbb{E}_{\mathbf{x} \sim p_\theta} \left[ \nabla_\theta E_\theta(\mathbf{x}) \right].
\end{aligned}$$

Substituting this back into the gradient of the loss:

$$\nabla_\theta \mathcal{L}_{\text{ML}} = \mathbb{E}_{\mathbf{x} \sim p_\mathcal{D}} \left[ \nabla_\theta E_\theta(\mathbf{x}) \right] - \mathbb{E}_{\mathbf{x} \sim p_\theta} \left[ \nabla_\theta E_\theta(\mathbf{x}) \right].$$

In practice, we denote the $\mathbf{x}^+$ the "positive" samples from the empirical data distribution $p_D$, and $\mathbf{x}^-$ the "negative" samples from the model:

$$\nabla_\theta \mathcal{L}_{\text{ML}} \approx \mathbb{E}_{\mathbf{x}^+ \sim p_\mathcal{D}} \left[ \nabla_\theta E_\theta(\mathbf{x}^+) \right] - \mathbb{E}_{\mathbf{x}^- \sim p_\theta} \left[ \nabla_\theta E_\theta(\mathbf{x}^-) \right].$$

## B.2 EBM training algorithm

To train our Energy-Based Models (EBMs), we follow the approach of [16]. Algo. 2 details the general training procedure:

---

**Algorithm 2:** Training Energy-Based Model using Langevin Dynamics

---

**Input:** Training dataset :$\mathcal{D}$, learning rate $\eta$, Replay Buffer $\mathcal{B}$, Langevin step size $\alpha$, noise scale $\sigma$, number of Langevin steps $L$

**while** Training **do**

    $\mathbf{x}^+ \sim \mathcal{D}$   sample from the dataset

    $\mathbf{x}^0 \sim \mathcal{B}$  # sample from a replay buffer with probability 95%

    ## Refine negative samples using Langevin dynamics

    **for** $t \leftarrow 1$ **to** $L$ **do**

      |  $\mathbf{x}^{t+1} \leftarrow \mathbf{x}^t - \alpha \nabla_{\mathbf{x}^t} E_\theta(\mathbf{x}^t) + \omega$   with $\omega \sim \mathcal{N}(0, \sigma)$

    $\mathbf{x}^- = \mathbf{x}^L$.detach()

    $\nabla_\theta \mathcal{L}_{\text{ML}} \approx \mathbb{E}_{\text{Batch}} \left[ \nabla_\theta E_\theta(\mathbf{x}_i^+) - \nabla_\theta E_\theta(\mathbf{x}_i^-) \right]$ ## Compute the ML loss

    $\mathcal{L}_{REG}(\theta) = \mathbb{E}_{\text{Batch}} \left[ \nabla_\theta E_\theta(\mathbf{x}_i^+)^2 + \nabla_\theta E_\theta(\mathbf{x}_i^-)^2 \right]$ ## Compute Regularization loss

    $\theta \leftarrow \theta - \eta \nabla_\theta \mathcal{L}_{\text{ML}} - \eta \nabla_\theta \mathcal{L}_{REG}$## update parameters with gradient descent

    $\mathcal{B} \leftarrow \mathcal{B} \cup \mathbf{x}^+$

---

In all experiments, we use $L = 100$ Langevin steps with step size $\alpha = 1$ and noise scale $\sigma = 10^{-2}$. The energy function is optimized using the Adam optimizer [94] with a learning rate of $\eta = 10^{-4}$. In addition to the maximum likelihood (ML) loss, we include a regularization term that encourages the energy values to remain close to zero, a technique shown to be effective in prior work [16].

We observed that training can be unstable, particularly for high-dimensional datasets. We attribute this instability to the lack of gradient supervision: the loss is not backpropagated through the Langevin dynamics to reduce memory usage. To mitigate this, we introduce a small Denoising Score Matching (DSM) loss—only for the AFHQ dataset—which provides weak supervision of the energy gradient. This additional regularization loss is similar to the DSM loss in [95]. We found this trick to strongly improve stability without degrading the performance.

The energy network architecture is adapted to the complexity of each dataset. Full details are provided in Appendix D.2, E.3, and F.2. Following Li et al. [95], we design the output layer of the energy function to take a quadratic form.

## B.3 Other training procedure in literature

EBM can also be trained by minimizing the so-called *Stein discrepancy* [96], Denoising Score Matching [97], Sliced Score Matching [98], Noise Contrastive Estimation [99]. A related objective to contrastive divergence is *energy discrepancy* [100]. We refer the reader [20], for a complete review of the different methods to train EBMs.

## C Riemannian Metrics

### C.1 Calibration

We normalize each metric using calibration coefficients $\alpha$ and $\beta$, with two goals: (i) ensuring that the Riemannian metric averages to the identity matrix $\mathbf{I}$ on the manifold, and (ii) aligning the overall scale of all metrics to allow fair comparisons. Here are more details on the calibration procedure:

First, we randomly sample data pairs $(\mathbf{x}_0, \mathbf{x}_1)$ from the dataset $\mathcal{D}$ (it corresponds to the geodesics endpoints) and generate linear interpolations between them using:

$$\mathbf{x}_t = (1-t)\mathbf{x}_0 + t\mathbf{x}_1 \tag{11}$$

Second, we define two sets of samples: $\mathcal{S}_{\mathcal{M}}$, which contains the endpoints $\mathbf{x}_0$ and $\mathbf{x}1$ *lying on the data manifold*, and $\mathcal{S}_{\bar{\mathcal{M}}}$, which contains the midpoints at $t = \frac{1}{2}$. These sets are then used to estimate the calibration coefficients $\alpha$ and $\beta$:

$$\mathbf{G}(\mathbf{x}) = \alpha\, \mathbf{h}(\boldsymbol{x}) + \beta \quad \text{s.t} \quad \begin{cases} \alpha = \dfrac{g_{\max} - g_{\min}}{\frac{1}{|\mathcal{S}_{\bar{\mathcal{M}}}|}\sum_{\mathbf{x}\in\mathcal{S}_{\bar{\mathcal{M}}}} h(\mathbf{x}) - \frac{1}{|\mathcal{S}_{\mathcal{M}}|}\sum_{\mathbf{x}\in\mathcal{S}_{\mathcal{M}}} h(\mathbf{x})} \\ \beta = g_{\min} - \alpha \cdot \frac{1}{|\mathcal{S}_{\mathcal{M}}|}\sum_{\mathbf{x}\in\mathcal{S}_{\mathcal{M}}} h(\mathbf{x}) \end{cases}$$

$$\mathbf{G}(\mathbf{x}) = (\alpha\, \mathbf{h}(\boldsymbol{x}) + \beta)^{-1} \quad \text{s.t} \quad \begin{cases} \alpha = \dfrac{1/g_{\max} - 1/g_{\min}}{\frac{1}{|\mathcal{S}_{\bar{\mathcal{M}}}|}\sum_{\mathbf{x}\in\mathcal{S}_{\bar{\mathcal{M}}}} h(\mathbf{x}) - \frac{1}{|\mathcal{S}_{\mathcal{M}}|}\sum_{\mathbf{x}\in\mathcal{S}_{\mathcal{M}}} h(\mathbf{x})} \\ \beta = \dfrac{1}{g_{\min}} - \alpha \cdot \frac{1}{|\mathcal{S}_{\mathcal{M}}|}\sum_{\mathbf{x}\in\mathcal{S}_{\mathcal{M}}} h(\mathbf{x}) \end{cases}$$

This calibration strategy adjusts the metric based on both on-manifold and off-manifold regions. It ensures that all metrics operate within a comparable dynamic range and promotes a useful geometric prior: lower metric values near the data manifold and higher values farther away. As a result, geodesics are encouraged to stay close to high-density areas, aligning the geometry with the data distribution.

### C.2 LAND metric

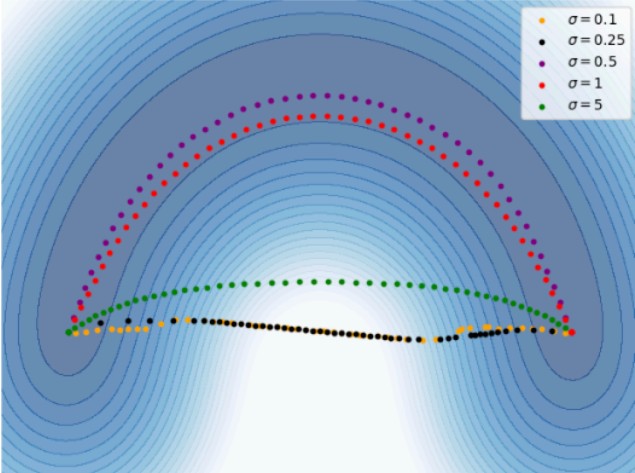

Figure 7: **Effect of the bandwidth** $\sigma$ on the geodesics obtained with the LAND metric. Here we have explored five $\sigma$ values ($\sigma \in \{0.1, 0.25, 0.5, 1, 5\}$). We observed that $\sigma$ has a major impact on the shape of the geodesics.

We remind the land metric formula (see Eq. 8):

$$\mathbf{G}_{\text{LAND}}(\mathbf{x}) = (\alpha\, \text{diag}(\mathbf{h}(\mathbf{x})) + \beta\mathbf{I})^{-1} \text{ s.t } h^{(j)}(\mathbf{x}) = \sum_{i=1}^{N}(x_i^{(j)} - x^{(j)})^2 \exp\left(-\frac{||\mathbf{x}-\mathbf{x_i}||^2}{2\sigma^2}\right) \tag{12}$$

This metric is highly sensitive to the choice of the $\sigma$ parameter, which controls the "locality" of the metric. A small $\sigma$ results in a very local metric that is strongly influenced by nearby points, while a large $\sigma$ smooths the metric by averaging over a wider region. This directly affects the trade-off between how closely geodesics follow the data manifold and how smooth or stable they are. In practice, we observe that $\sigma$ has a major impact on the shape of the geodesics, as shown in Fig.7, confirming earlier findings by[11]. To illustrate this, we plot geodesics for five different values of $\sigma$ ($\sigma \in \{0.1, 0.25, 0.5, 1, 5\}$) and find that they closely follow the data manifold only within a narrow range, particularly around $\sigma = 0.5$.

## C.3   RBF metric

We first remind the RBF formula :

$$\mathbf{G}_{\text{RBF}}(\mathbf{x}) = (\alpha \cdot h(\mathbf{x}) + \beta)^{-1} \cdot \mathbf{I}, \quad h(\mathbf{x}) = \sum_{k=1}^{K} w_k \exp\left(-0.5 \cdot \lambda_k \|\mathbf{x} - \hat{\mathbf{x}}_k\|^2\right).$$

In the equation, the $\{\hat{\mathbf{x}}\}_{i=1}^{K}$ are centroids evaluated using a K-Means algorithm. Following [13], the bandwidth ($\lambda_k$) using the inter-distance to prototype (see Eq. 13):

$$\lambda_k = \frac{1}{2}\left(\frac{\kappa}{2K}\sum_{k=1}^{K}\|\mathbf{x} - \hat{\mathbf{x}}_\mathbf{k}\|\right)^{-2} \tag{13}$$

The bandwidth, $\lambda_k$, controls the spatial extent of each radial basis function. In Eq. 13, $\kappa$ is a tunable hyperparameter controlling how concentrated or spread out the RBFs are. Intuitively, a larger $\kappa$ results in narrower kernels (stronger locality) while a smaller one yields wider coverage. This trade-off is explored via hyperparameter search. The weights $w_k$ modulate the relative contribution of each RBF to the resulting scalar field. These weights are optimized to ensure that $h(x)$ remains close to 1 on the training data, using the following loss:

$$\mathcal{L}(\mathbf{w}) = \sum_{n=1}^{N} \|1 - h(\mathbf{x_i})\|^2 \tag{14}$$

This encourages the RBF combination to approximate a constant value (here, 1) across the data distribution, ensuring consistency and stability of the field on the manifold.

In Fig. 8, we evaluate how the number of centroids $K$ affects the shape of the geodesics. The results show that geodesics are highly sensitive to this parameter. When $K$ is too small, the geodesics fail to follow the data manifold accurately. Conversely, when $K$ is too large, the trajectories become overly sinuous—passing through many centroids that are not necessarily aligned with the true manifold.

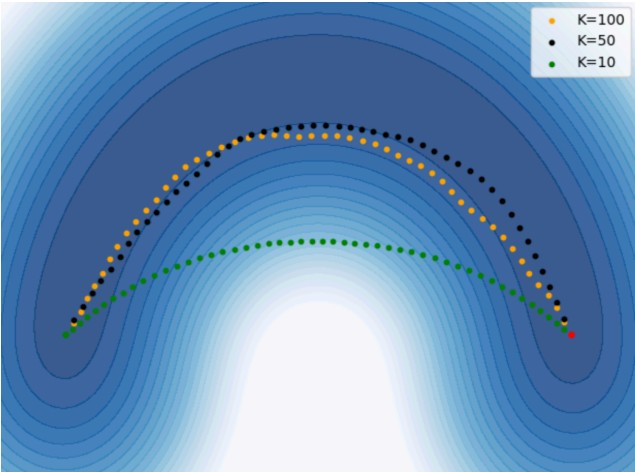

Figure 8: **Effect of the number of centroids** $K$ on the geodesics obtained with the RBF metric ($K \in \{10, 50, 100\}$). We observed that $K$ has a major impact on the shape of the geodesics.

## D  Experimental details on the Circular Mixture of Gaussian datasets

### D.1  Datasets

To design our toy datasets, we have used a mixture of K (2D) Gaussians. Specifically, $K = 200$ in all our datasets. The resulting probability distribution is therefore:

$$p(\boldsymbol{x}) = \sum_{k=1}^{K} \pi_k \, \mathcal{N}(\boldsymbol{x} \mid \boldsymbol{\mu}_k, \mathbf{I}), \tag{15}$$

where $\mathcal{N}(\boldsymbol{x} \mid \boldsymbol{\mu}_k, \mathbf{I})$ denotes a 2D isotropic Gaussian centered at $\boldsymbol{\mu}_k$. Here, $\mathbf{I}$ is the identity matrix of size $2 \times 2$. In both datasets, the centers of the Gaussians are uniformly positioned along a semi-circle or Radius $R$ (here $R = 8$). Specifically, the centers are given by:

$$\boldsymbol{\mu}_k = R \cdot \begin{bmatrix} \cos(\theta_k) \\ \sin(\theta_k) \end{bmatrix} \quad \text{with} \quad \theta_k = \frac{k}{K} \cdot \pi, \quad k = 0, \dots, K-1. \tag{16}$$

The only difference between the Uniform Circular Gaussian (UCG) dataset and the Weighted Circular Gaussian dataset (WCG) is the weighting coefficient $\{\pi_k\}_{k=1}^{K}$

**Uniform Circular Gaussian dataset.** Here all the weights are similar and equal to $1/K$. As a result, the energy landscape forms a semi-circular basin with constant depth (see contour plot of Fig. 2a for an illustration of the energy landscape).

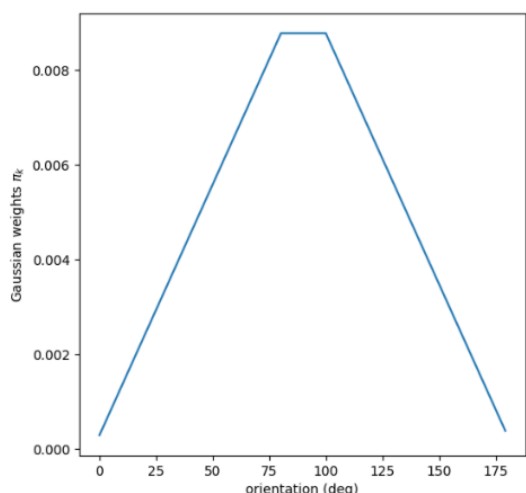

Figure 9: **Profile of the Gaussians weights** $\pi_k$

**Weighted Circular Gaussian dataset.** In this setting, the mixture weights vary, concentrating the distribution toward the center of the arc. The weights are symmetric with respect to the horizontal axis, producing an energy landscape with a semi-circular shape and slopes symmetric around the arc's midpoint (see the contour plot in Fig.2c). Fig.9 shows the weights $\pi_k$ as a function of orientation, with all weights summing to 1. This setup generates a curved, non-uniform density with higher mass near the center of curvature (i.e., at 90 degrees), allowing us to introduce a controlled curvature in the data manifold and assess how well different metrics capture it.

### D.2  Neural networks architectures and Hyperparameters on the Circular Mixture of Gaussian Dataset

Here, we describe the architecture of the energy function (see Table 2), the interpolant network (see Table 3), and the hyperparameters used for the $\mathbf{G_{LAND}}$ and $\mathbf{G_{RBF}}$ metrics. Note that the architectures and settings are the same for both the UCG and WCG datasets.

**Energy-Based Model**  Table 2 summarizes the architecture used for the energy function of the EBM. The output is designed to follow a quadratic form, similar to the approach in [95], which we found improves performance across all datasets. To assess whether the EBM successfully learns the target distribution, we visualize the learned energy landscapes for both the UCG and WCG datasets (see Fig. 10a and Fig. 10b, respectively). For reference, we also include the ground-truth energy landscapes of the target distributions (see Fig. 10c and Fig. 10d for UCG and WCG, respectively). We observe that the EBM accurately captures the overall shape of the energy landscape for both distributions. However, in the WCG dataset, the true energy spans a broader range than the EBM's learned energy. This discrepancy is partially corrected by the normalization procedure described in Appendix C.1.

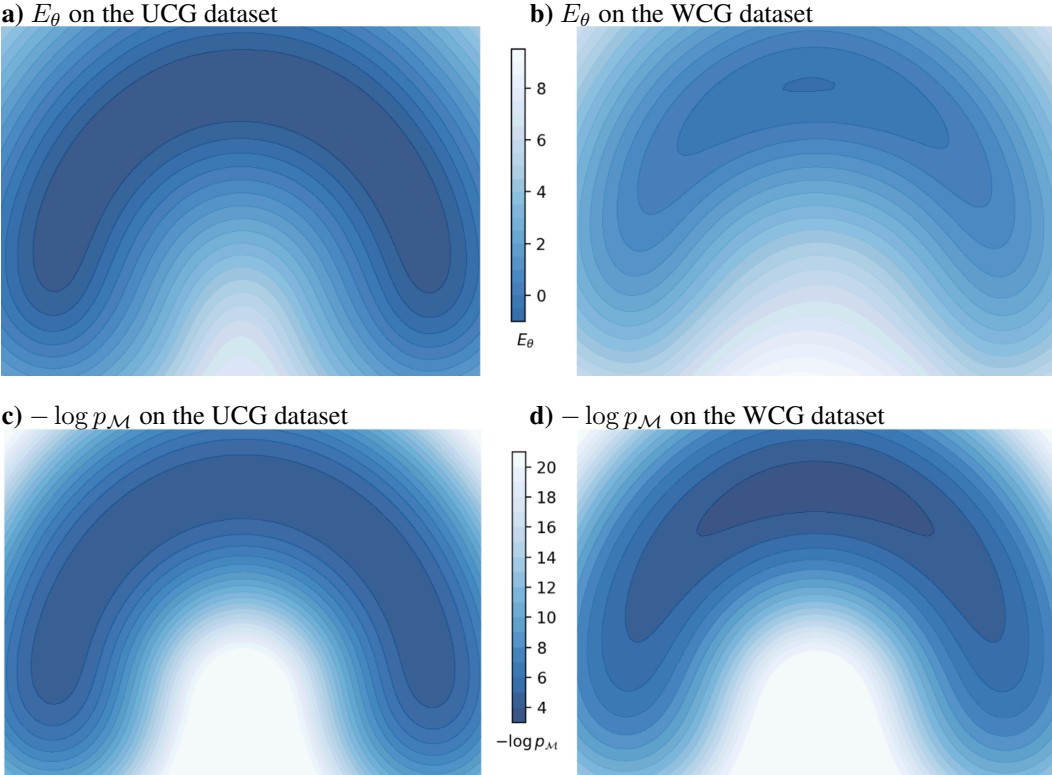

Figure 10: **Energy Landscape on the UCG and WCG datasets. (a, b)** shows the energy landscape learned by the EBMs on the UCG and WCG datasets, respectively. **(c, d)** shows the true energy landscape (i.e., $-\log p_{\mathcal{M}}$) on the UCG and WCG datasets, respectively.

**Interpolant Network** Table 3 summarizes the architecture used for the interpolant network (i.e., $\varphi_{t,\eta}$ in Algo. 1 and Eq. 3). For all datasets, we use an autoencoder-like architecture for the interpolant, following a similar approach to [24].

| Nb. Layers | Layer type |
|---|---|
| 1 | Linear (2, 32) SiLU |
| 4 | Linear (32, 32) SiLU |
| 1 | Linear (32, 32) |
| 1 | Three output heads: Linear (32, 1) for $f_1$ Linear (32, 1) for $f_2$ Linear (32, 1) for $f_3$ |
| output | $f_1(x) \cdot f_2(x) + f_3(x^2)$ |

Table 2: MLP architecture of the energy function on both UCG and WCG datasets.

| NB. Layers | Layer type |
|---|---|
| 1 | Linear (3, 32) SiLU |
| 1 | Linear (32, 64) SiLU |
| 1 | Linear (64, 64) SiLU |
| 1 | Linear (64, 32) SiLU |
| 1 | Linear (32, 3) |

Table 3: MLP architecture of the interpolant network $\varphi_{t,\eta}$ for WCG dataset.

**LAND metric** We performed a hyperparameter search to tune the $\sigma$ parameter. We found that $\sigma = 1$ yielded the best performance. Parameters are similar for both UCG and WCG.

**RBF metric** We conducted a hyperparameter search to tune both the number of centroids $K$ and the scaling factor $\kappa$. The best results were obtained with $K = 30$ and $\kappa = 1$. Parameters are similar for both UCG and WCG.

### D.3 Quantitative evaluation with error bars

In Fig. 11, we report the same quantitative results as in Fig. 2, now including 2-$\sigma$ error bars. The standard deviation $\sigma$ is computed over evaluation metrics, each averaged on a different set of randomly sampled trajectories (five sets in total).

**a)** Geodesics evaluation on UCG

| Metric | $p_{\mathcal{M}}(\gamma^\star)$ ($\uparrow$) | RMSE ($\downarrow$) |
|---|---|---|
| $G_{E_{\mathcal{M}}}$ | $0.79 \pm 0.02$ | - |
| $G_{1/p_{\mathcal{M}}}$ | $0.77 \pm 0.04$ | - |
| $G_{E_\theta}$ | $\mathbf{0.78} \pm 0.03$ | $0.12 \pm 0.02$ |
| $G_{1/p_\theta}$ | $0.73 \pm 0.01$ | $\mathbf{0.10} \pm 0.03$ |
| $G_{LAND}$ | $0.60 \pm 0.07$ | $0.38 \pm 0.05$ |
| $G_{RBF}$ | $0.61 \pm 0.06$ | $0.39 \pm 0.1$ |

**b)** Geodesics evaluation on WCG

| Metric | $p_{\mathcal{M}}(\gamma^\star)$ ($\uparrow$) | RMSE ($\downarrow$) |
|---|---|---|
| $G_{E_{\mathcal{M}}}$ | $0.67 \pm 0.05$ | - |
| $G_{1/p_{\mathcal{M}}}$ | $0.73 \pm 0.07$ | - |
| $G_{E_\theta}$ | $\mathbf{0.67} \pm 0.06$ | $0.18 \pm 0.07$ |
| $G_{1/p_\theta}$ | $0.67 \pm 0.09$ | $\mathbf{0.14} \pm 0.06$ |
| $G_{LAND}$ | $0.65 \pm 0.11$ | $0.34 \pm 0.05$ |
| $G_{RBF}$ | $0.47 \pm 0.14$ | $2.2 \pm 0.1$ |

Figure 11: **Quantitative evaluation of the geodesics on the UCG and WCG datasets.** We report (i) the accumulated probability along the geodesic (the higher the better) and ii) RMSE between each geodesic and its corresponding baseline (the lower the better). Values after the $\pm$ sign indicate the 2-$\sigma$ error.

# E    Experimental details on the Rotated Character Dataset

## E.1    Datasets

The Rotated Character Datasets consist of 7 printed characters (5, G, F, P, J, 7, 2), represented as black-and-white images of size $32 \times 32$. These characters were selected for two main reasons: (i) they are commonly used in psychophysics experiments [63], and (ii) they are asymmetric and visually distinct, which helps avoid ambiguities in the resulting geodesic trajectories. Fig. 12 shows all characters in their unrotated form.

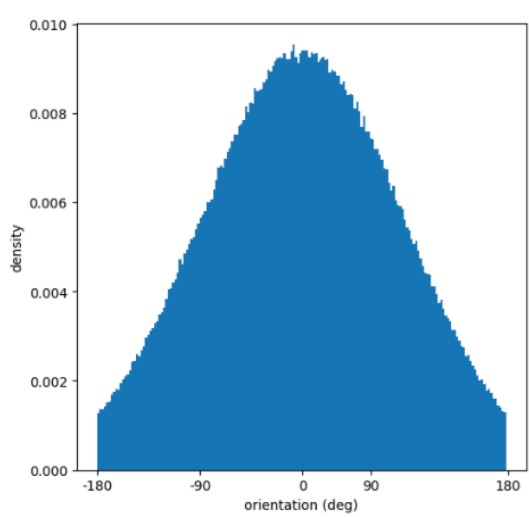

Figure 12: **Original (non-rotated) samples from the Rotated Character Dataset**

The only difference between the Uniform Rotated Character (URC) and Biased Rotated Character (BRC) datasets lies in the distribution of character orientations.

**Uniform Rotated Character (URC)**    In this setting, character orientations are sampled uniformly across the full range of $[-179°, 180°]$, using a one-degree step. This ensures that each possible orientation within this interval is equally likely. Importantly, the distribution is consistent across all characters, meaning that each character appears with the same uniform spread of rotations.

**Biased Rotated Character (BRC)**    Here, orientations follow a truncated Gaussian distribution centered at $0°$, designed to mimic natural rotation statistics (see Fig. 13). Unlike the Mixture of Gaussian datasets, we do not have access to a closed-form expression for the underlying distribution $p_\mathcal{M}$, but we do control its empirical form.

Figure 13: **Distribution of orientation for the BRC dataset**

This setup introduces a controlled curvature in the data manifold, allowing us to assess how well different metrics adhere to it.

## E.2    Architecture and algorithm of the Triplet Loss autoencoder

We computed geodesics in the latent space of an autoencoder trained with a Triplet Loss [101]. This approach is motivated by the fact that image space is inherently non-Euclidean, making it poorly suited for defining meaningful distances. In contrast, the latent space of our autoencoder is explicitly regularized so that Euclidean distances correspond to differences in orientation. By

---

**Algorithm 3:** Autoencoder with Triplet regularization

**Input:** Dataset $\mathcal{D} = \{(\mathbf{x}_i, \theta_i)\}$, Encoder $E_\phi$, Decoder $D_\psi$

**while** training **do**

    Sample bath of triplet B=$(\mathbf{x}_a, \mathbf{x}_p, \mathbf{x}_n)$ from $\mathcal{D}$

    # Same character; $\theta_p$ close to $\theta_a$, $\theta_n$ farther

    $\mathbf{z}_a = E_\phi(\mathbf{x}_a), \quad \mathbf{z}_p = E_\phi(\mathbf{x}_p), \quad \mathbf{z}_n = E_\phi(\mathbf{x}_n)$

    $\mathcal{L}_{\text{rec}} = \|D_\psi(\mathbf{z}_a) - \mathbf{x}_a\|^2$

    $\Delta\theta_p = |\theta_a - \theta_p|, \quad \Delta\theta_n = |\theta_a - \theta_n|$

    $\mathcal{L}_{\text{T}} = \mathbb{E}_{\text{B}}\left((\|\mathbf{z}_a - \mathbf{z}_p\| - \alpha\Delta\theta_p)^2 + (\|\mathbf{z}_a - \mathbf{z}_n\| - \alpha\Delta\theta_n)^2\right)$

    $\mathcal{L}_{\text{total}} = \mathcal{L}_{\text{rec}} + \lambda \cdot \mathcal{L}_{\text{T}}$

    Update $(\phi, \psi)$ using gradient $\nabla\mathcal{L}_{\text{total}}$

---

treating the latent space as the ambient space for geodesic computation, we align with the assumption that the data manifold is embedded in an Euclidian Manifold. The training procedure is described in

Algo. 3, and the encoder and decoder architectures—based on the Regularized Autoencoder (RAE) framework [102]—are detailed in Table 4 and Table 5, respectively.

We trained the model using the Adam optimizer [94] with a learning rate of $1 \times 10^{-4}$ and a batch size of 128. In Algorithm 3, we set $\alpha = 1$ and $\lambda = 0.1$. For the architecture, the number of input features (i.e., the number of channels in the first convolutional layer) was set to $F = 128$. In Table 4 and Table 5, the notation "Conv2D($n_c$, $n_f$, 3, 1)" refers to a convolutional layer with $n_c$ input channels, $n_f$ output channels, a kernel size of 3, and padding of 1. Similarly, "ConvTr2D" denotes a transposed convolution. The RAE blocks are modules introduced in [102], referred to here as RaeBlockDown and RaeBlockUp, and are used for efficient downsampling and upsampling, respectively.

| Nb. Layers | Layer Type |
|---|---|
| 1 | Conv2d $(1, F, 3, 1)$ |
| 1 | RaeBlockDown $(F, 2F)$ 
 ReLU |
| 1 | Conv2d $(2F, 2F, 3, 1)$ |
| 1 | RaeBlockDown $(2F, 4F)$ 
 ReLU |
| 1 | Conv2d $(4F, 4F, 3, 1)$ 
 ReLU |
| 1 | Linear $(4F * 8 * 8, z)$ |

Table 4: Encoder architecture of the autoencoder. $F$ is the number of features ($F = 128$), and z is the size of the latent space ($z = 64$).

| Nb. Layers | Layer Type |
|---|---|
| 1 | ConvTr2d $(z, 4F, 8, 0)$ 
 ReLU |
| 1 | Conv2d $(4F, 4F, 3, 1)$ |
| 1 | RaeBlockUp $(4F, 2F)$ 
 ReLU |
| 1 | Conv2d $(2F, 2F, 3, 1)$ |
| 1 | RaeBlockUp $(2F, F)$ 
 ReLU |
| 1 | Conv2d $(F, F, 3, 1)$ |
| 1 | Conv2d $(F, 1, 4, 1)$ 
 Tanh |

Table 5: Decoder architecture of the autoencoder. $F$ is the number of features ($F = 128$), and z is the size of the latent space ($z = 64$).

### E.3 Architecture of the energy function and the interpolant network on the Rotated Character Dataset

The architecture of the energy function used in the EBM is shown in Table 6, and the architecture of the interpolant network is provided in Table 7. These architectures are used for both the URC and BRC datasets. The EBM was trained using the procedure described in Algorithm 2, and Fig.14 shows samples generated by the EBM at the end of training. All EBM training hyperparameters match those described in SectionB.2. For both the EBM and interpolant training, we use a batch size of 128. The interpolant network is optimized with Adam, using a learning rate of $1 \times 10^{-4}$.

| Nb. Layers | Layer Type |
|---|---|
| 1 | Linear (64, 128) 
 SiLU |
| 1 | Linear (128, 512) 
 SiLU |
| 6 | Linear (512, 512) 
 SiLU |
| 1 | Linear (512, 64) |
| 1 | Three output heads: 
 Linear (64, 1) for $f_1$ 
 Linear (64, 1) for $f_2$ 
 Linear (64, 1) for $f_3$ |
| Output | $f_1(x) \cdot f_2(x) + f_3(x^2)$ |

Table 6: Archiecture of the EBM energy function on both URC and BRC datasets

| Nb. Layers | Layer Type |
|---|---|
| 1 | Linear (64*3, 128) 
 SiLU |
| 1 | Linear (128, 128) 
 SiLU |
| 1 | Linear 128, 128) 
 SiLU |
| 1 | Linear (128, 128) 
 SiLU |
| 1 | Linear (128, 128) 
 SiLU |
| 1 | Linear 128, 64) 
 SiLU |

Table 7: Architecture of the interpolant network used on the URC and BRC dataset.

### E.4 Hyperparameters of the LAND and RBF metric

**LAND metric**  We performed a hyperparameter search to tune the $\sigma$ parameter. We found that $\sigma = 0.4$ yielded the best performance. Parameters are similar for both the URC and BRC datasets.

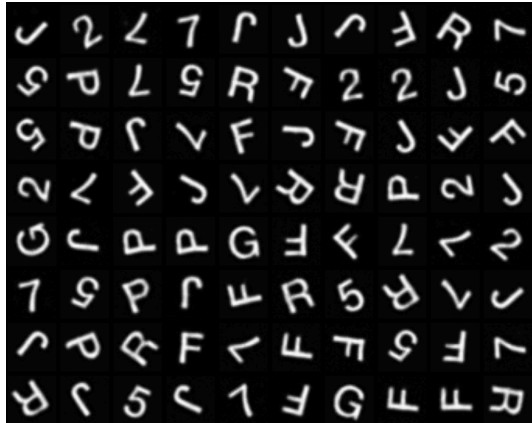

Figure 14: **Samples from the EBM train on URC.** These samples are generated by applying Langevin dynamics to the energy function learned by the EBM.

**RBF metric** We conducted a hyperparameter search to tune both the number of centroids $K$ and the scaling factor $\kappa$. The best results were obtained with $K = 300$ and $\kappa = 0.75$. Parameters are similar for both the URC and BRC datasets.

## E.5 Additional geodesics

**URC dataset:** In Fig. 15 we show additional geodesics on the URC dataset.

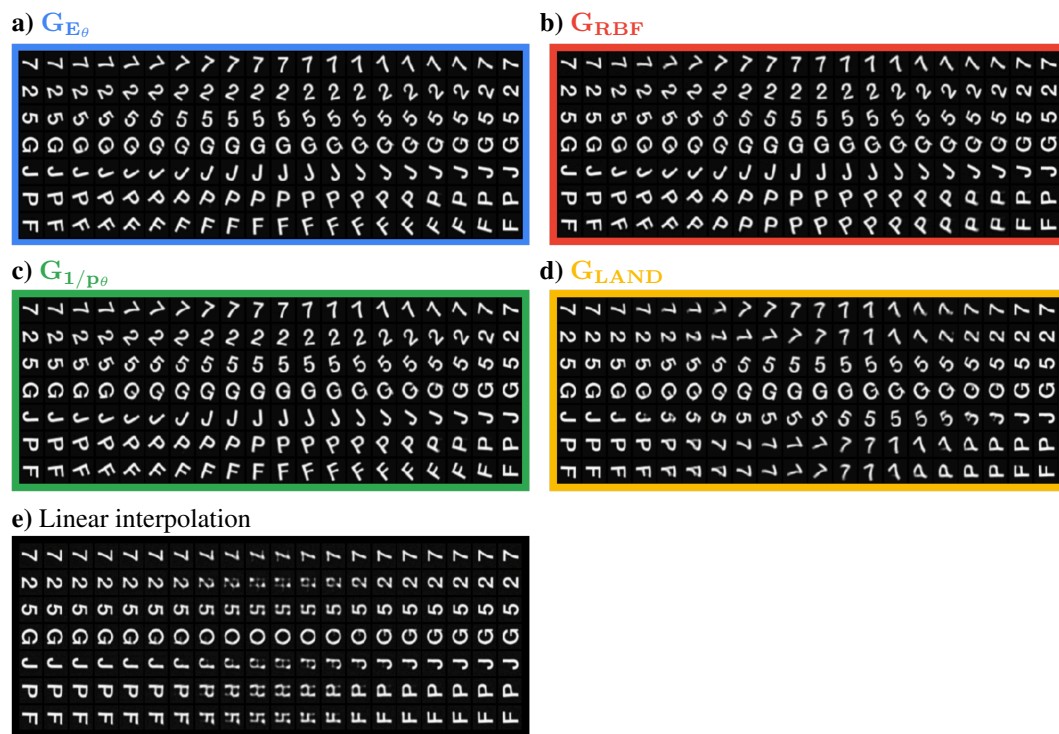

Figure 15: **Geodesics on the URC dataset.** Geodesics are computed using four different metrics: **a)** $\mathbf{G}_{\mathbf{E}_\theta}$, **b)** $\mathbf{G}_{\mathbf{RBF}}$, **c)** $\mathbf{G}_{\mathbf{1}/\mathbf{p}_\theta}$, **d)** $\mathbf{G}_{\mathbf{LAND}}$. For comparison, a simple linear interpolation is shown in **e)**. The trajectory are computed in the latent space of the autoencoder and projected into pixel space for visualization. Each trajectory is subsampled at 20 time steps for clarity.

**BRC dataset:**  In Fig. 16 we show additional geodesics on the BRC dataset.

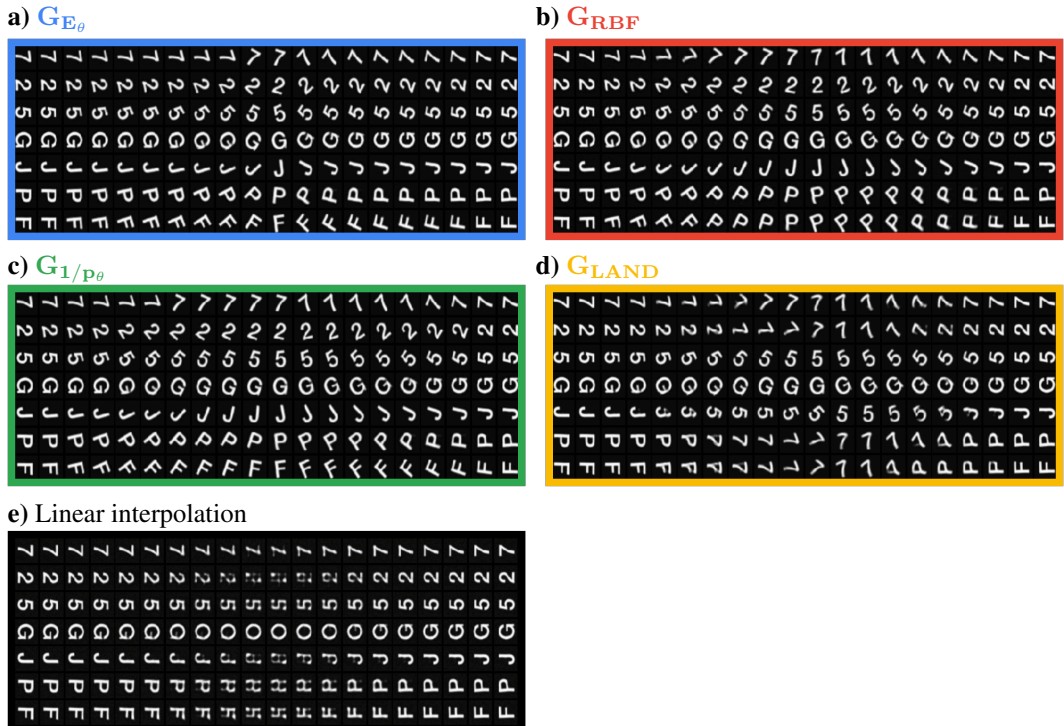

Figure 16: **Geodesics on the BRC dataset.** Geodesics are computed using four different metrics: **a)** $\mathbf{G_{E_\theta}}$, **b)** $\mathbf{G_{RBF}}$, **c)** $\mathbf{G_{1/p_\theta}}$, **d)** $\mathbf{G_{LAND}}$. For comparison, a simple linear interpolation is shown in **e)**. The trajectory are computed in the latent space of the autoencoder and projected into pixel space for visualization. Each trajectory is subsampled at 20 time steps for clarity.

## E.6  Quantitative evaluation with error bars

In Table. 8, we report the same quantitative results as in Fig. 4, now including 2-$\sigma$ error bars. The standard deviation $\sigma$ is computed over evaluation metrics, each averaged on a different set of randomly sampled trajectories (five sets in total).

| Metric | $\mathcal{D}$-RMSE ($\downarrow$) | $\gamma^\star$-RMSE ($\downarrow$) |
|---|---|---|
| linear interp. | $2.96 \pm 0.42$ | $3.52 \pm 0.21$ |
| $\mathbf{G_{E_\theta}}$ | $\mathbf{0.11 \pm 0.01}$ | $\mathbf{0.40 \pm 0.03}$ |
| $\mathbf{G_{1/p_\theta}}$ | $0.14 \pm 0.02$ | $0.44 \pm 0.07$ |
| $\mathbf{G_{LAND}}$ | $0.66 \pm 0.12$ | $2.39 \pm 0.51$ |
| $\mathbf{G_{RBF}}$ | $0.36 \pm 0.06$ | $0.86 \pm 0.17$ |

Table 8: **Quantitative evaluation on the URC dataset with the** $2\sigma$ **error.** Quantitative evaluation using two metrics: (i) $\mathcal{D}$-RMSE, which measures proximity to the dataset manifold, and (ii) $\gamma$-RMSE, which measures the deviation from an ideal smooth rotation. Values after the $\pm$ sign indicate the 2-$\sigma$ error.

# F  Experimental details on the Rotated Character Dataset

## F.1  Dataset

In this section, we conduct experiments on the Animal Faces High-Quality (AFHQ) dataset introduced by [59]. The full dataset contains 15,000 images across three categories: cats, dogs, and wild animals. For our experiments, we restrict the dataset to the cat and dog classes only, each comprising approximately 5,000 images. This choice avoids introducing curvature in the data manifold that could arise from the relatively small number of samples in the wild animal category. All images are cropped, aligned, and have a resolution of 512×512 pixels. AFHQ is widely used for image-to-image translation and style transfer, and its diversity in pose, breed, and appearance makes it well-suited for smooth interpolation tasks. See Fig. 17 for example images from the AFHQ dataset.

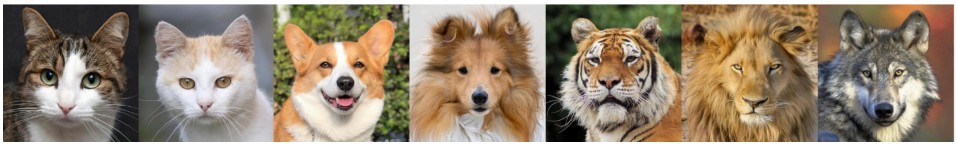

Figure 17: **Samples from the AFHQ dataset [59]**

For the experiments in this section, we compute geodesics in the latent space of a pretrained Variational Autoencoder (VAE). Specifically, we use the VAE from Stable Diffusion V1 [18]. The latent representations have a spatial size of $4 \times 16 \times 16$.

## F.2  Architecture of the energy function and the interpolant network on the AFHQ dataset

**Energy Function:**  The architecture used for the energy function is detailed in Table 9. We set the number of input channels to $n_c = 4$, matching the dimensionality of the latent representation, and use $F = 256$ feature channels in the first convolutional layer. The network follows a simple sequence of downsampling convolutional layers, which we found to yield more stable training than ResNet-style architectures. The EBM is trained using Algorithm 2, with the same hyperparameters as in Section B.2. To further improve training stability, we add a denoising score matching regularization term and use a cosine learning rate scheduler.

| Nb. Layers | Layer Type |
|:---:|:---:|
| 1 | Conv2d $(n_c, F, 3, 1, 1)$
SiLU |
| 1 | Conv2d $(F, F, 3, 1, 1)$
SiLU |
| 1 | Conv2d $(F, 2F, 4, 2, 1)$
SiLU |
| 1 | Conv2d $(2F, 2F, 3, 1, 1)$
SiLU |
| 1 | Conv2d $(2F, 4F, 4, 2, 1)$
SiLU |
| 1 | Conv2d $(4F, 4F, 3, 1, 1)$
SiLU |
| 1 | Conv2d $(4F, 8F, 4, 2, 1)$
SiLU |
| 1 | Conv2d $(8F, 1, 2, 1, 0)$: for $f_1$ |
| 1 | Conv2d $(8F, 1, 2, 1, 0)$: for $f_2$ |
| 1 | Conv2d $(8F, 1, 2, 1, 0)$: for $f_3$ |
| Output | $f_1(x) \cdot f_2(x) + f_3(x^2)$ |

Table 9: **Architecture of the energy function**. $F$ denotes the base number of feature channels, and $n_c$ is the number of input channels. The final energy is computed using three parallel output heads. The notation Conv2d($n_c$, $n_f$, $k$, $s$, $p$) refers to a 2D convolutional layer with $n_c$ input channels, $n_f$ output channels, a kernel size of $k$, stride $s$, and padding $p$.

In Fig. 18, we show randomly selected samples generated by the EBM after training. The Fréchet Inception Distance (FID) of the model is measured to be 9.89.

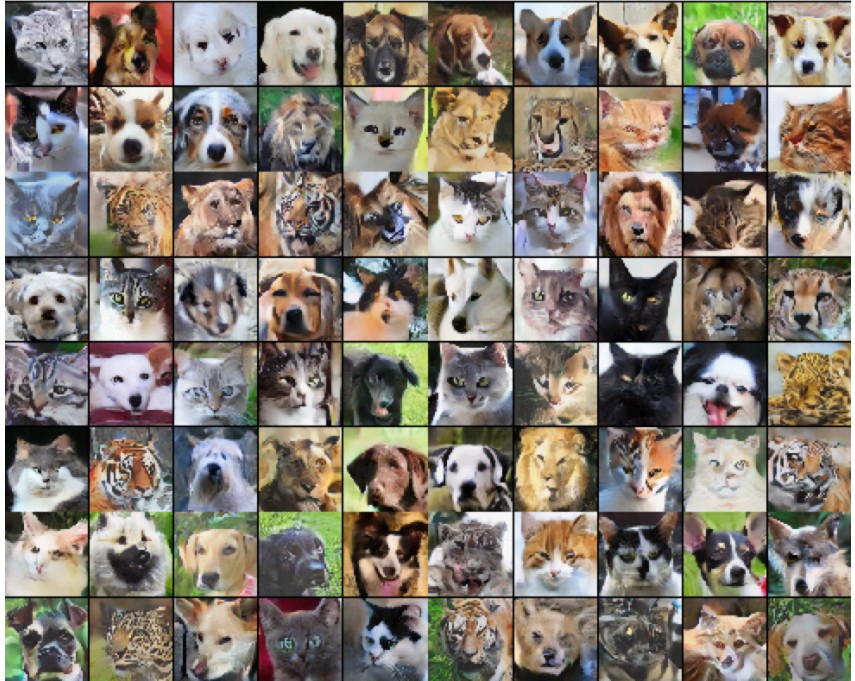

Figure 18: **Samples from the EBM trained on the AFHQ dataset.** These samples are generated by applying Langevin dynamics to the energy function learned by the EBM.

**Interpolant Network:**  We use the U-Net architecture from [103], following the same hyperparameter settings.

### F.3  Hyperparameters of the LAND and RBF metric

**LAND metric**  We performed a hyperparameter search to tune the $\sigma$ parameter. We found that $\sigma = 10$ yielded the best performance.

**RBF metric**  We conducted a hyperparameter search to tune both the number of centroids $K$ and the scaling factor $\kappa$. The best results were obtained with $K = 1000$ and $\kappa = 3$.

### F.4  FIDs with error bars

In Table. 10, we include 2-$\sigma$ error bars. The standard deviation $\sigma$ is computed over different sets of randomly sampled trajectories (five sets in total).

| Metric | FID ($\downarrow$) |
|---|---|
| Linear interp. | $42.47 \pm 3.17$ |
| Slerp interp. | $32.67 \pm 2.33$ |
| $\mathbf{G_{E_\theta}}$ | $20.79 \pm 2.17$ |
| $\mathbf{G_{1/p_\theta}}$ | $\mathbf{16.47} \pm 1.04$ |
| $\mathbf{G_{LAND}}$ | $39.17 \pm 3.63$ |
| $\mathbf{G_{RBF}}$ | $37.98 \pm 2.46$ |

Table 10: **FID along geodesics for different Riemannian metrics**. FID is computed at each trajectory point to assess on-manifold alignment. Values after the $\pm$ sign indicate the 2-$\sigma$ error.

## F.5 Additional geodesics on AFHQ

**Riemanian metric:** $\mathbf{G}_{1/\mathbf{p}_\theta}$

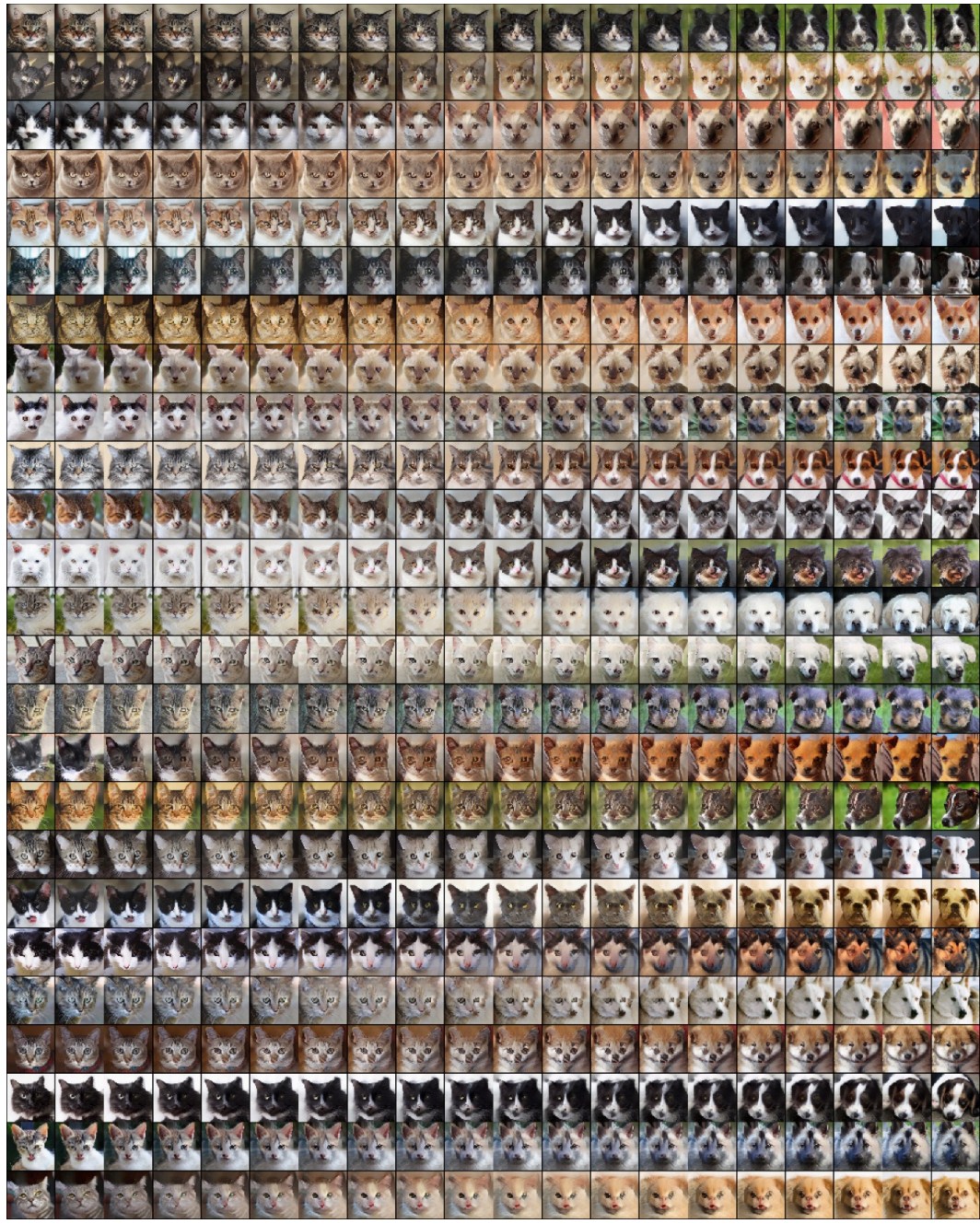

Figure 19: **Geodesics on the AFHQ dataset using $\mathbf{G}_{1/\mathbf{p}_\theta}$.** Each row shows a geodesic in latent space between a randomly sampled cat image (start point) and a dog image (end point). Columns correspond to time steps along each geodesic, from left (start) to right (end). Images are obtained by decoding the latent representations back into pixel space.

**Riemanian metric: $\mathbf{G}_{\mathbf{E}_\theta}$**

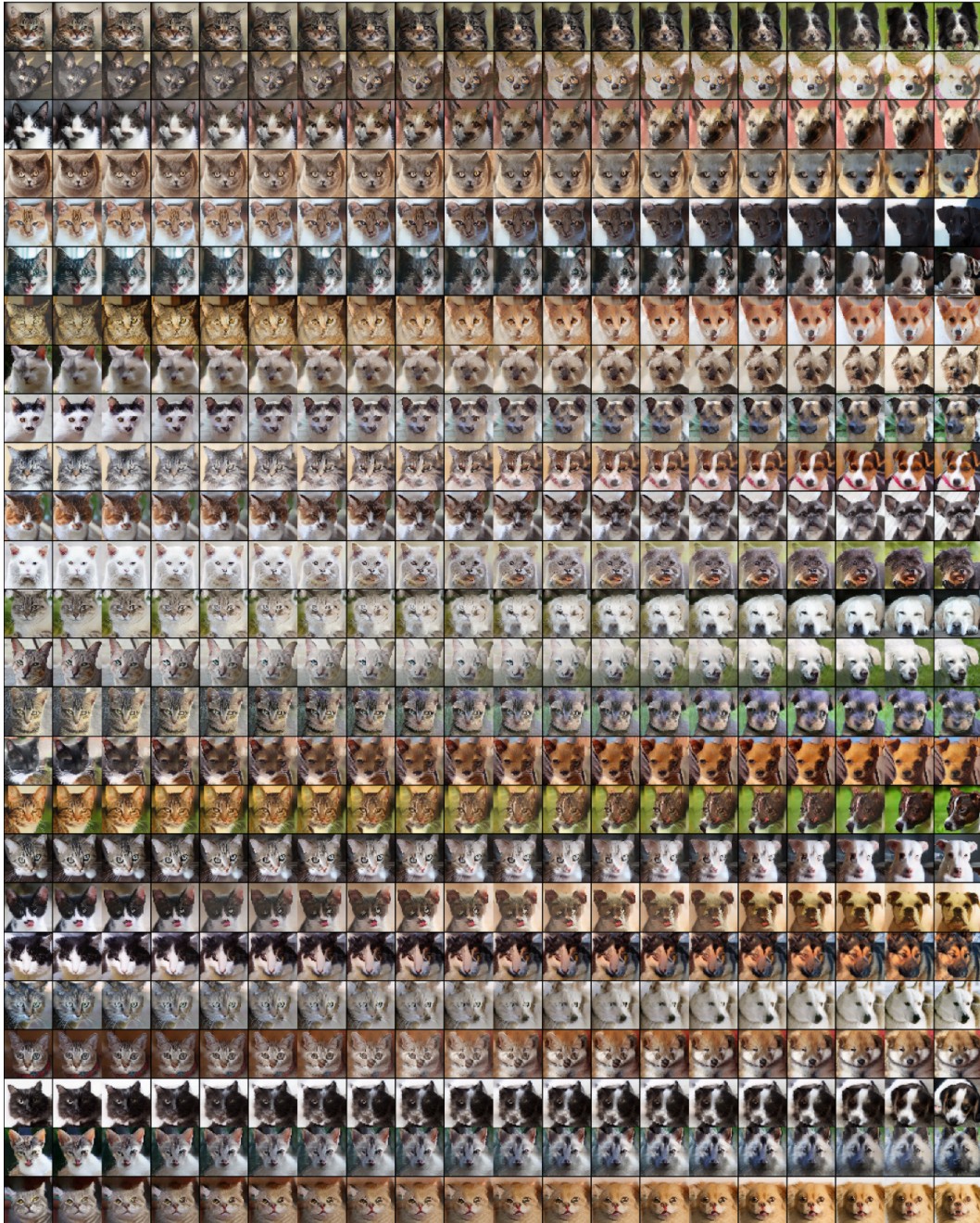

Figure 20: **Geodesics on the AFHQ dataset using $\mathbf{G}_{\mathbf{E}_\theta}$.** Each row shows a geodesic in latent space between a randomly sampled cat image (start point) and a dog image (end point). Columns correspond to time steps along each geodesic, from left (start) to right (end). Images are obtained by decoding the latent representations back into pixel space.

**Riemanian metric: $\mathbf{G_{RBF}}$**

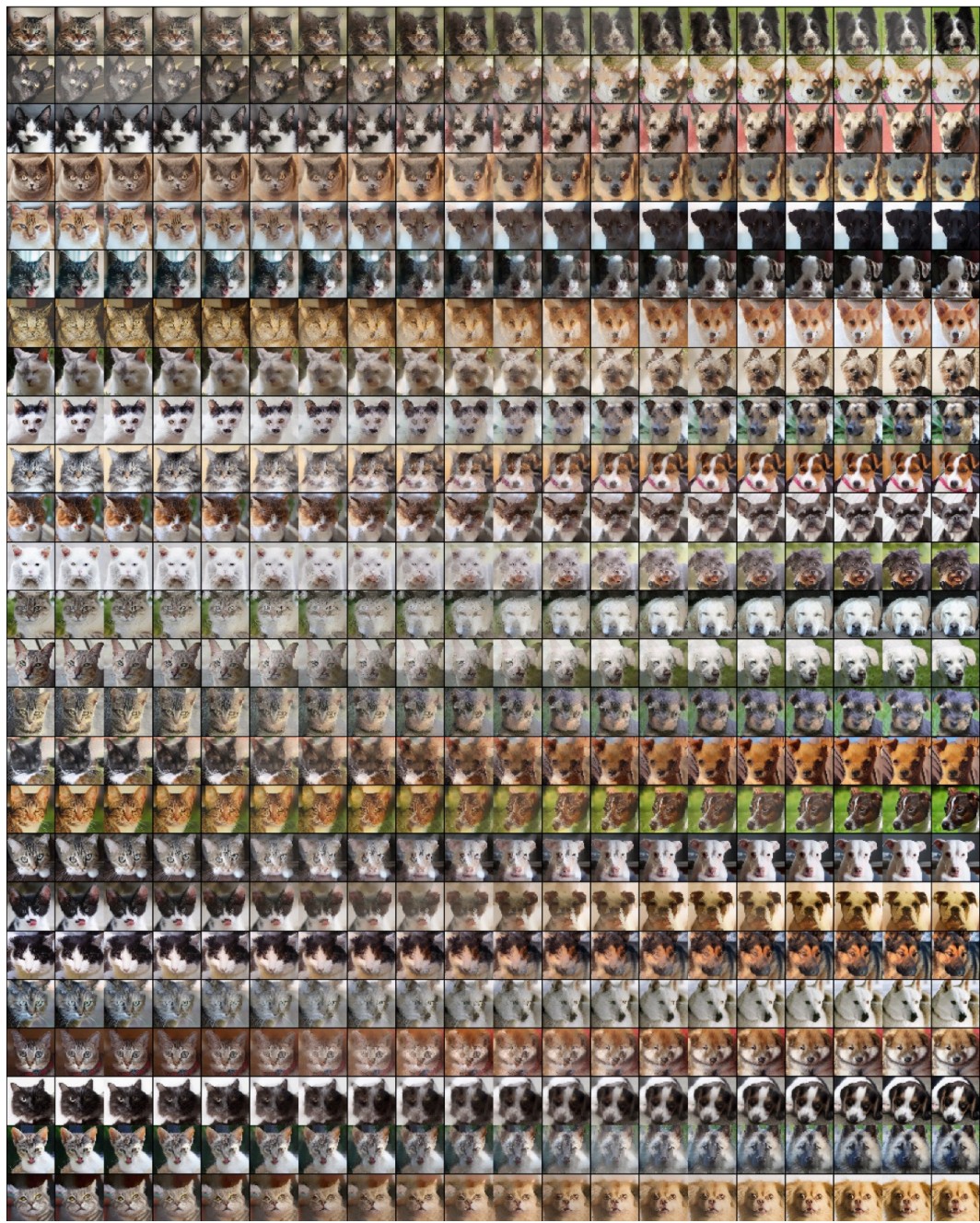

Figure 21: **Geodesics on the AFHQ dataset using $\mathbf{G_{RBF}}$.** Each row shows a geodesic in latent space between a randomly sampled cat image (start point) and a dog image (end point). Columns correspond to time steps along each geodesic, from left (start) to right (end). Images are obtained by decoding the latent representations back into pixel space.

**Riemanian metric:** $\text{G}_{\text{LAND}}$

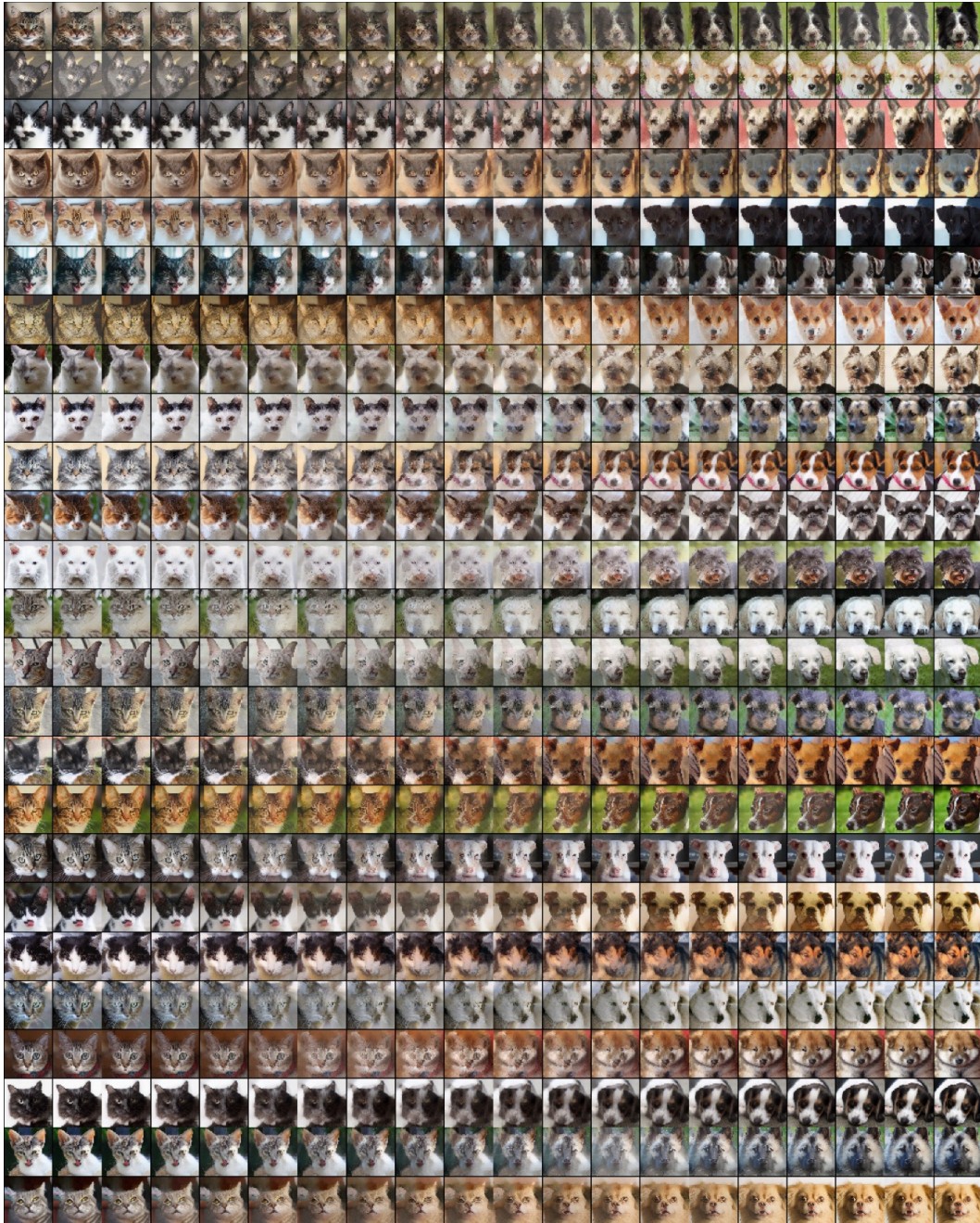

Figure 22: **Geodesics on the AFHQ dataset using** $\text{G}_{\text{LAND}}$**.** Each row shows a geodesic in latent space between a randomly sampled cat image (start point) and a dog image (end point). Columns correspond to time steps along each geodesic, from left (start) to right (end). Images are obtained by decoding the latent representations back into pixel space.

**Linear interpolation**

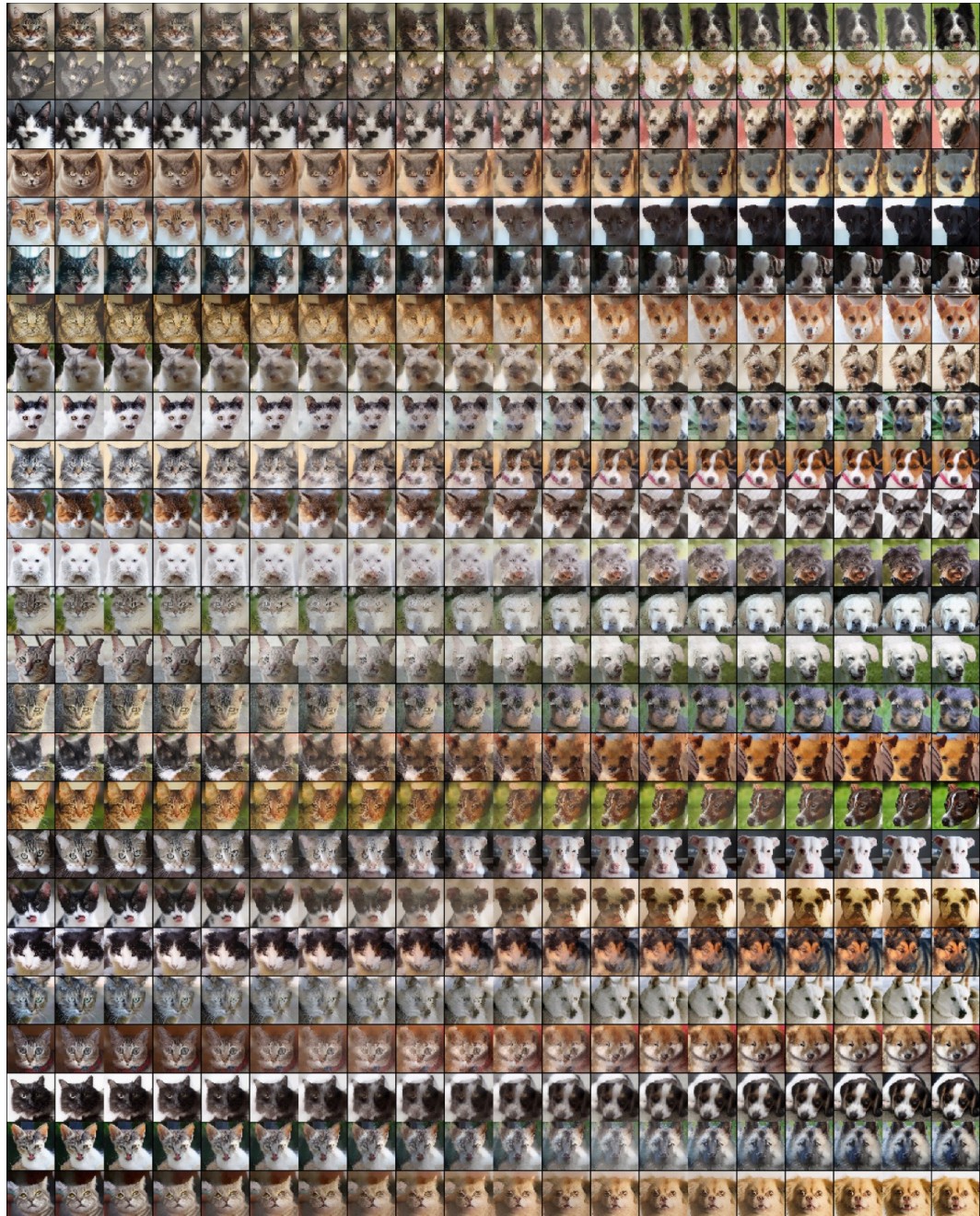

Figure 23: **Linear interpolation on the AFHQ dataset.** Each row shows an interpolation in latent space between a randomly sampled cat image (start point) and a dog image (end point). Columns correspond to time steps along each interpolation, from left (start) to right (end). Images are obtained by decoding the latent representations back into pixel space.

**Slerp interpolation**

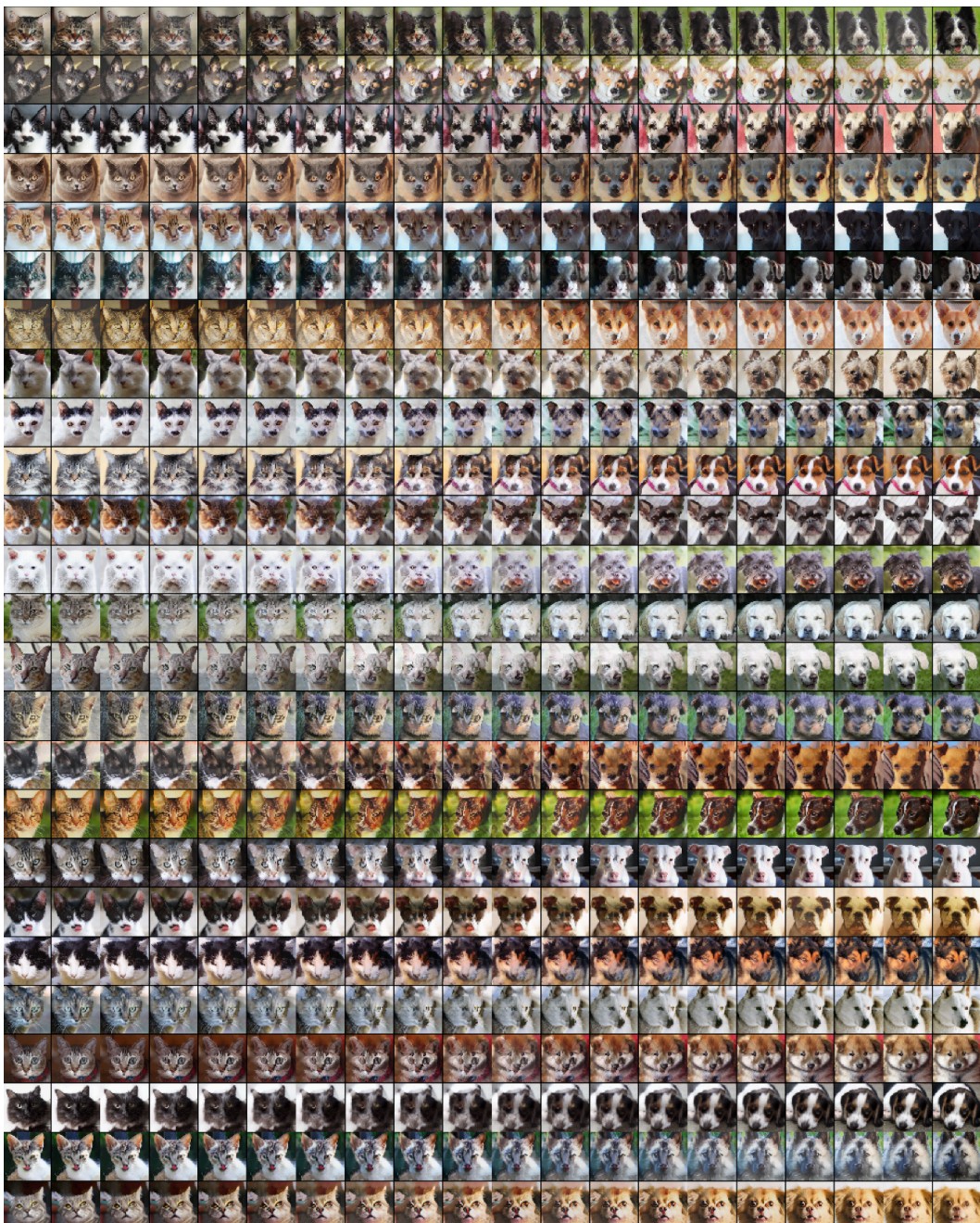

Figure 24: **Spherical interpolation (Slerp) on the AFHQ dataset.** Each row shows an interpolation in latent space between a randomly sampled cat image (start point) and a dog image (end point). Columns correspond to time steps along each interpolation, from left (start) to right (end). Images are obtained by decoding the latent representations back into pixel space.

### F.6 About the Spherical interpolation

Given two points $\mathbf{x}_0, \mathbf{x}_1 \in \mathbb{R}^D$ lying on the unit hypersphere (i.e., $\|\mathbf{x}_0\| = \|\mathbf{x}_1\| = 1$), the spherical interpolation between them is defined as:

$$\text{slerp}(t; \mathbf{x}_0, \mathbf{x}_1) = \frac{\sin((1-t)\theta)}{\sin\theta}\mathbf{x}_0 + \frac{\sin(t\theta)}{\sin\theta}\mathbf{x}_1, \quad t \in [0, 1],$$

where $\theta$ is the angle between $\mathbf{x}_0$ and $\mathbf{x}_1$, given by:

$$\theta = \arccos\left(\frac{\langle \mathbf{x}_0, \mathbf{x}_1 \rangle}{\|\mathbf{x}_0\| \|\mathbf{x}_1\|}\right).$$

In practice, when interpolating latent codes from a Variational Autoencoder (VAE), the latent vectors $\mathbf{x}_0$ and $\mathbf{x}_1$ are typically drawn from a standard normal prior and do not lie on the unit sphere. To apply slerp, we first normalize the vectors:

$$\tilde{\mathbf{x}}_0 = \frac{\mathbf{x}_0}{\|\mathbf{x}_0\|}, \qquad \tilde{\mathbf{x}}_1 = \frac{\mathbf{x}_1}{\|\mathbf{x}_1\|},$$

and compute $\theta$ as:

$$\theta = \arccos\left(\langle \tilde{\mathbf{x}}_0, \tilde{\mathbf{x}}_1 \rangle\right).$$

This interpolation method, introduced by [60], has proven particularly effective for interpolating in the latent space of VAEs. The intuition behind its success is that it implicitly assumes the data manifold lies on a hypersphere. While this may seem restrictive, the assumption is reasonable in practice. In a VAE, each latent coordinate $x_i$ is drawn from a standard Normal distribution: $x_i \sim \mathcal{N}(0, 1)$ $(1 < i < D)$. As a result, the squared norm, $||\mathbf{x}||^2 = \sum_{i=1}^{D} x_i^2$ follows a chi-squared distribution with $D$ degree of freedom. This distribution is known to concentrate tightly around $D$, effectively placing most latent codes near the surface of a hypersphere. To validate this empirically, we visualize the distribution of $||\mathbf{x}||^2$ for all latent codes of the AFHQ dataset (see Fig. 25). We observe that this distribution is concentrated on $D$—$D = 1024$ for VAE with latent space of size $4 \times 16 \times 16$.

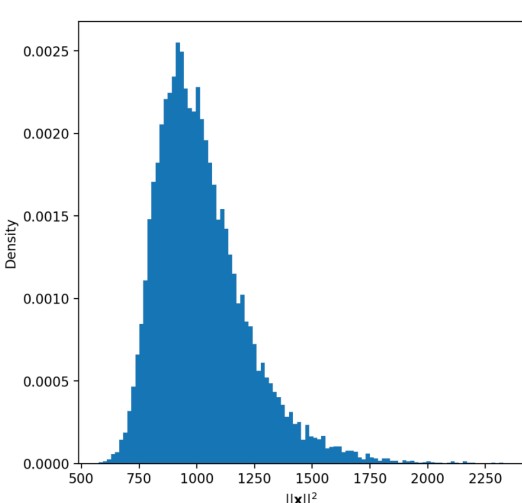

Figure 25: **Distribution of $||\mathbf{x}||^2$ on the AFHQ dataset**

To conclude, slerp interpolation is well-suited for VAE latent spaces because it aligns with their underlying geometric structure.

### F.7 Physical interpretation

We refer the reader to [104] or [105] for a detailed background on differential geometry.

**Geodesic equation.** Assume that the manifold $\mathcal{M}$ is the ambient $D$-dimensional Euclidean space ($\mathcal{M} = \mathbb{R}^D$). We equipped the manifold $\mathcal{M}$ with a conformal Riemannian metric $\mathbf{G}(\mathbf{x}) = \dfrac{1}{p(\mathbf{x})} \cdot \mathbf{I}$, with $p$ the probability density of the data, and $\mathbf{I}$ the identity matrix of $\mathbb{R}^{D \times D}$. Let $\boldsymbol{\gamma}(t)$ be a geodesic (i.e. $\gamma : [0,1] \to \mathbb{R}^D$). We denote the instantaneous speed of the geodesic at time $t$, $\dot{\boldsymbol{\gamma}}(t)$, and its acceleration $\ddot{\boldsymbol{\gamma}}(t)$. Said otherwise, $\dot{\boldsymbol{\gamma}}(t)$ and $\ddot{\boldsymbol{\gamma}}(t)$ denote $\frac{\partial \gamma}{\partial t}(t)$ and $\frac{\partial^2 \gamma}{\partial t^2}(t)$ respectively.

The geodesic equation is the 2nd-order ODE written as:

$$\ddot{\boldsymbol{\gamma}}^k(t) + \sum_{i,j} \Gamma_{i,j}^k(\boldsymbol{\gamma}(t)) \cdot \dot{\boldsymbol{\gamma}}^i(t) \cdot \dot{\boldsymbol{\gamma}}^j(t) = 0 \tag{17}$$

In Eq.17, $\ddot{\boldsymbol{\gamma}}^k(t)$ and $\dot{\boldsymbol{\gamma}}^k(t)$ denotes the k-th coordinate of $\ddot{\boldsymbol{\gamma}}(t)$ and $\dot{\boldsymbol{\gamma}}(t)$, respectively (here $1 < k < D$). $\Gamma_{i,j}^k$ are the Christoffel symbols, they are derived from the Riemannian metric and encode how it bends and curves the space. $\Gamma_{i,j}^k$ tells how much the change in direction in the $i$-th and $j$-th coordinate causes acceleration in the $k$-th coordinate ($1 < i, j, k < D$). Said differently, $i, j$ refer to the coordinate direction along which the particule is moving, and $k$ refers to the coordinate direction where the motion causes effect (i.e. curvature induces acceleration).

**Christoffel symbols for conformal metric.** The Christoffel symbols for a conform metric $\mathbf{G}(\mathbf{x}) = \lambda(\mathbf{x}) \cdot \mathbf{I}$ (with $\lambda$ a scalar function):

$$\Gamma_{ij}^k(\mathbf{x}) = \frac{1}{2\lambda(\mathbf{x})} \left( \delta_{j,k}\, \partial_i \lambda(\mathbf{x}) + \delta_{i,k}\, \partial_j \lambda(\mathbf{x}) - \delta_{ij}\, \partial_k \lambda(\mathbf{x}) \right) \tag{18}$$

In Eq. 18, $\partial_i \lambda(\mathbf{x}) = \dfrac{\partial \lambda(\mathbf{x})}{\partial x^i}$ (i.e. the partial derivative of $\lambda(\mathbf{x})$ with respect to the $i$-th coordinate), and $\delta_{j,k}$ is the Kronecker symbol ( $\delta_{j,k} = 1$ if $j = k$ and $\delta_{j,k} = 0$ otherwise). If one plugs Eq. 18 in the right hand side of Eq. 17:

$$\sum_{i,j} \Gamma_{ij}^k\big(\boldsymbol{\gamma}(t)\big)\, \dot{\boldsymbol{\gamma}}^i(t)\, \dot{\boldsymbol{\gamma}}^j(t) = \frac{1}{2\lambda\big(\boldsymbol{\gamma}(t)\big)} \cdot \Bigg[ \sum_i \partial_i \lambda\big(\boldsymbol{\gamma}(t)\big)\, \dot{\boldsymbol{\gamma}}^i(t)\, \dot{\boldsymbol{\gamma}}^k(t)$$

$$+ \sum_j \partial_j \lambda\big(\boldsymbol{\gamma}(t)\big)\, \dot{\boldsymbol{\gamma}}^k(t)\, \dot{\boldsymbol{\gamma}}^j(t)$$

$$- \sum_i \partial_k \lambda\big(\boldsymbol{\gamma}(t)\big)\, \dot{\boldsymbol{\gamma}}^i(t)^2 \Bigg]$$

$$= \frac{1}{2\lambda\big(\boldsymbol{\gamma}(t)\big)} \cdot \Big[ 2\dot{\boldsymbol{\gamma}}^k(t)\big\langle \nabla\lambda\big(\boldsymbol{\gamma}(t)\big), \dot{\boldsymbol{\gamma}}(t)\big\rangle - \partial_k \lambda\big(\boldsymbol{\gamma}(t)\big)\, \|\dot{\boldsymbol{\gamma}}(t)\|^2 \Big], \tag{19}$$

where $\langle \cdot, \cdot \rangle$ and $\| \cdot \|$ are the usual *Euclidean* inner product and norms, respectively.

So Eq. 17, becomes :

$$\ddot{\boldsymbol{\gamma}}^k(t) = -\frac{\dot{\boldsymbol{\gamma}}^k(t)}{\lambda\big(\boldsymbol{\gamma}(t)\big)}\big\langle \nabla\lambda\big(\boldsymbol{\gamma}(t)\big), \dot{\boldsymbol{\gamma}}(t)\big\rangle + \frac{1}{2\lambda\big(\boldsymbol{\gamma}(t)\big)} \partial_k \lambda\big(\boldsymbol{\gamma}(t)\big)\, \|\dot{\boldsymbol{\gamma}}(t)\|^2 \tag{20}$$

**Pulling everything together.** If one plugs our definition of the Riemannian metric (i.e. $\lambda\big(\boldsymbol{\gamma}(t)\big) = \dfrac{1}{p\big(\boldsymbol{\gamma}(t)\big)}$, and therefore $\dfrac{\nabla\lambda\big(\boldsymbol{\gamma}(t)\big)}{\lambda\big(\boldsymbol{\gamma}(t)\big)} = -\nabla \log p\big(\boldsymbol{\gamma}(t)\big)$), Eq. 20 becomes:

$$\ddot{\boldsymbol{\gamma}}(t) = \big\langle \nabla_\gamma \log p(\boldsymbol{\gamma}(t)), \dot{\boldsymbol{\gamma}}(t)\big\rangle \cdot \dot{\boldsymbol{\gamma}}(t) - \frac{1}{2}\|\dot{\boldsymbol{\gamma}}(t)\|^2 \cdot \nabla_\gamma \log p(\boldsymbol{\gamma}(t)) \tag{21}$$

Eq. 21 is similar in form to Newton's second law. The acceleration of a particle (of unit mass) is governed by a velocity-dependent force built from the Stein Score (i.e. $\nabla_\gamma \log p(\boldsymbol{\gamma}(t))$). More speficically:

- $\langle \nabla \log p(\boldsymbol{\gamma}(t)), \dot{\boldsymbol{\gamma}}(t) \rangle \cdot \dot{\boldsymbol{\gamma}}(t)$ describes a "force" aligned with the particle velocity direction. This term acts like an anisotropic drag or propulsion term: i) it speeds up the particle when it goes toward a high-density region and ii) it slows down the particle going the other way.
- $-\frac{1}{2}\|\dot{\boldsymbol{\gamma}}(t)\|^2 \cdot \nabla \log p(\boldsymbol{\gamma}(t))$ is a force in the direction of the stein score (pointing toward low density regions). It behaves like a repulsive force, pushing the particle toward areas with low probability. The faster the particle moves, the stronger the force.

The "force" seems to depend on the velocity $\dot{\boldsymbol{\gamma}}(t)$, which is typical of inertial forces (i.e, forces that depend on a given frame). This is an artifact from the affine parametrization of the geodesic, which ensures constant speed along the trajectory.

**Newtonian formalism.** Note that the variable $t$ in previous equations is the geometrical "time". This variable $t$ stems from the affine parametrization (e.g. see Eq. 3) and is not related to the physical time. To make Eq. 21 compatible with the "physical" time, denoted $s$, one can consider the following change of variable:

$$\frac{\partial s}{\partial t}(t) = p(\boldsymbol{\gamma}(t)) \quad \text{or equivalently} \quad \frac{\partial t}{\partial s}(s) = \frac{1}{p(\boldsymbol{\gamma}(t(s)))} \tag{22}$$

This change of variable implies that when moving through space according to arc-length s, the geometric time $t$ runs more slowly in low-density regions and faster in high-density ones. This change of variable is particularly handy to interpret Eq. 21 as Newtonian motion. Let's therefore consider the following reparametrization: $\boldsymbol{\gamma}(t(s)) = \mathbf{x}(s)$, where $\mathbf{x}$ is the new trajectory parametrized by the physical time $s$. So:

$$\dot{\boldsymbol{\gamma}}(t) = \frac{\partial}{\partial t}\boldsymbol{\gamma}(t) = \frac{\partial}{\partial t}\mathbf{x}(s(t)) = \frac{\partial \mathbf{x}}{\partial s} \cdot \frac{\partial s}{\partial t} = \dot{\mathbf{x}}(s) \cdot p(\mathbf{x}(s)) \tag{23}$$

$$\ddot{\boldsymbol{\gamma}}(t) = \frac{\partial}{\partial t}\left(\dot{\mathbf{x}}(s) \cdot p(\mathbf{x}(s))\right) = \left(\frac{\partial}{\partial s}\left(\dot{\mathbf{x}}(s) \cdot p(\mathbf{x}(s))\right)\right) \cdot \frac{\partial s}{\partial t}$$

$$= \left(\ddot{\mathbf{x}}(s) \cdot p(\mathbf{x}(s)) + \langle \nabla p(\mathbf{x}(s)), \dot{\mathbf{x}}(s) \rangle \cdot \dot{\mathbf{x}}(s)\right) \cdot p(\mathbf{x}(s))$$

$$= p(\mathbf{x})^2 \ddot{\mathbf{x}} + p(\mathbf{x}) \langle \nabla p(\mathbf{x}), \dot{\mathbf{x}} \rangle \dot{\mathbf{x}} \tag{24}$$

Now plugging Eq. 24 and Eq. 23 in Eq. 21:

$$p(\mathbf{x})^2 \ddot{\mathbf{x}} + p(\mathbf{x}) \langle \nabla p(\mathbf{x}), \dot{\mathbf{x}} \rangle \dot{\mathbf{x}}$$

$$= \langle \nabla \log p(\mathbf{x}), \dot{\boldsymbol{\gamma}}(t) \rangle \cdot \dot{\boldsymbol{\gamma}}(t) - \frac{1}{2}\|\dot{\boldsymbol{\gamma}}(t)\|^2 \cdot \nabla \log p(\mathbf{x})$$

$$= \langle \nabla \log p(\mathbf{x}), p(\mathbf{x})\dot{\mathbf{x}} \rangle \cdot (p(\mathbf{x})\dot{\mathbf{x}}) - \frac{1}{2}\|p(\mathbf{x})\dot{\mathbf{x}}\|^2 \cdot \nabla \log p(\mathbf{x})$$

$$= p(\mathbf{x})^2 \langle \nabla \log p(\mathbf{x}), \dot{\mathbf{x}} \rangle \dot{\mathbf{x}} - \frac{1}{2}p(\mathbf{x})^2\|\dot{\mathbf{x}}\|^2 \cdot \nabla \log p(\mathbf{x})$$

$$\Rightarrow \quad \ddot{\mathbf{x}} = -\frac{1}{2}\|\dot{\mathbf{x}}\|^2 \underbrace{\nabla \log p(\mathbf{x})}_{\text{Stein score}}.$$

This equation can be interpreted through Newton's second law: it describes the motion of a particle $\mathbf{x}$ following a geodesic in the Riemanannian manifold $\left(\mathcal{M}, \frac{1}{p(\mathbf{x})}\right)$, where $p(\mathbf{x})$ denotes the data density. The particle experiences a force $-\frac{1}{2}\|\dot{\mathbf{x}}\|^2 \nabla \log p(\mathbf{x})$, pushing away from regions of high probability. The term $\|\mathbf{x}\|^2$ modulates the forces magnitude and plays a role analogous to momentum, strengthening the pull when the particle moves quickly. While this is not a literal physical system— here the particle is a data point, and has no mass, it provides a useful analogy for understanding the dynamics of trajectories shaped by data geometry.

# G  Limitations

While our approach provides a promising framework for deriving Riemannian metrics from EBMs, several limitations should be acknowledged:

- First, we restrict our study to conformal metrics, which uniformly scale the identity matrix and thus cannot capture directional (anisotropic) structure in the data manifold. While this simplifies optimization, it limits expressivity in settings where geometry varies across directions—something more expressive, score-based metrics may help resolve.

- Second, our method relies on pretrained EBMs that assign meaningful energy values across the entire space. Training such models is challenging in high-dimensional settings due to the computational cost of sampling (e.g., Langevin dynamics), and performance can degrade if the energy landscape is poorly shaped or overfitted.

- Third, although we demonstrate improvements over strong baselines, our evaluation of geodesic quality remains largely indirect—relying on alignment with proxy measures (e.g., density, rotation smoothness, FID). In complex datasets like natural images, the absence of ground-truth geometry makes rigorous evaluation difficult.

- Fourth, our approach assumes that the data distribution is adequately captured by the EBM, yet in practice, misestimation of density—especially in underrepresented regions—may distort the metric and lead to suboptimal paths.

- Finally, while we demonstrate promising results on several datasets, our experiments are constrained to pretrained generative models and fixed feature spaces (e.g., VAE latents), and generalizing to end-to-end learnable architectures remains unexplored.

Future work may address these limitations by developing scalable score-based metrics, improving EBM training stability, integrating richer evaluation protocols, and extending the framework to broader model classes and learning settings.

# H  Broader Impact

This work advances our understanding of data geometry by connecting generative modeling and Riemannian geometry, with potential implications across machine learning, neuroscience, and cognitive science. By enabling principled geodesic computation in high-dimensional spaces, our approach could support safer interpolation in generative models, improve motion planning in robotics, or inform models of human cognition. However, care should be taken when applying such methods to sensitive domains, as learned energy landscapes may inherit biases present in training data.

# I  Computational ressources

All experiments were conducted on NVIDIA RTX 3090 GPUs (32 GB memory). Training on the toy dataset was fast, with both the EBM and interpolant completing in a few minutes. For the Rotated Characters dataset, EBM training took 8 GPU hours and the interpolant 30 minutes. On the AFHQ dataset, training required 6 GPU days for the EBM and 24 GPU hours for the interpolant. Including extensive hyperparameter searches and trial-and-error development, the total compute usage amounted to approximately 123,000 GPU hours.

