# OpenReview forum: "Follow the Energy, Find the Path: Riemannian Metrics from Energy-Based Models"
_NeurIPS.cc/2025/Conference — NeurIPS 2025 poster_

### Official Review · Reviewer_z3Pm · 2025-06-28

**Clarity:** 4
**Significance:** 2
**Originality:** 3
**Rating:** 5
**Confidence:** 3

**Summary:**

This paper proposes two conformal Riemannian metrics defined from a pre-trained Energy-Based Model (EBM) for a given dataset. The metrics leverage the EBM's learned energy landscape to define a geometry where paths through high-density regions are shorter. Through experiments on datasets of increasing complexity, the authors provide evidence that their metrics produce geodesics that stay closer to the data manifold compared to established methods.

**Questions:**

My recommendation for this paper could be raised based on the authors' responses to the first two questions, which address the paper's scope and practical significance. The subsequent questions are for my interest and could hopefully help strengthen the paper further.

1.  **On the Restriction to Conformal Metrics:** The study is limited to conformal metrics. Could the authors please elaborate on the reasons for this choice? Was it primarily for computational benefits, or because estimating a full, non-conformal metric tensor from sparse high-dimensional data is fundamentally more difficult? Furthermore, could you provide some intuition on how one might expect geodesics to differ if they were computed with a non-conformal metric (e.g., say one derived from the Hessian of the energy function) versus the proposed conformal ones?

2.  **Envisioned Applications and Cost-Benefit Analysis:** Training an EBM is the most computationally expensive part of the proposed pipeline. Could the authors provide concrete examples of applications where the high fidelity of the resulting geodesics is a critical requirement? Specifically, what are some real-world problems where the benefits of these more accurate paths would warrant the significant upfront cost of training a large-scale energy-based model?

Thank you for considering these first two points, which are my main questions. The following are a few smaller suggestions that I hope will be helpful in further polishing the paper.

3.  **Clarification on Geodesic Behavior with Manifold Gaps:** The evaluation on rotated characters assumes an "ideal smooth rotation" as the ground truth. However, what is the principled or "correct" behavior if the data manifold has significant gaps (e.g., training data contains images at 0-30 degrees and 60-90 degrees, but nothing in between)? Should the geodesic cut across the empty latent space while preserving identity, or should it detour to a "cheaper" path, perhaps via another character's manifold? How do the proposed metrics behave in such a scenario?

4.  **Details on Figures 1 and 3:** The visualizations in Figure 1 are quite dense. A more detailed walkthrough of one example trajectory in the caption would improve readability. For instance, the caption notes the trajectories are projected into PCA space; a brief elaboration would be helpful. Similarly, for Figure 3, could you please clarify what "step" (x-axis) and "step size" (y-axis) represent? The y-axis appears to be the Euclidean norm of the velocity vector, but this could be explicitly stated. The paragraph explaining Figure 3 is also a little vague, and a more detailed explanation would strengthen the point being made.
5. **Question on Finite Difference Stability:** You make an interesting practical claim that using finite differences was "more stable and accurate" than full autodifferentiation for computing the geodesic path's velocity. Could you provide a little more detail or intuition as to why this was the case in your experiments?
6.  **Minor Suggestion:** Many of the plots are very effective in color, but their key details may be lost when printed in grayscale.

**Ethical Concerns:**

["NO or VERY MINOR ethics concerns only"]

**Final Justification:**

My initial score was borderline, based on concerns about the method's practical applicability. The authors' rebuttal successfully addressed these points by providing compelling use cases and discussing the restriction to conformal metrics. This has alleviated my main reservations.

While the desirability of the geodesic's behaviour across manifold gaps remains a nuanced, context-dependent issue, this is not a fundamental flaw of the method. The paper is original, clear, and makes a solid contribution to data-driven geometry. I have raised my score as a result and recommend acceptance.

**Limitations:**

Yes

**Quality:**

3

**Strengths And Weaknesses:**

**Quality and Clarity:**
The paper is exceptionally well-written, clear, and easy to follow. The authors do a good job of building intuition for complex topics in Riemannian geometry. For instance, the paragraph explaining why geodesics accelerate in high-density regions is insightful and helpful for readers less familiar with the topic. The experimental setup is convincing and showcases the method's capabilities.

**Originality and Significance:**
The primary strength of this work lies in its originality. The proposal to derive Riemannian metrics directly from pretrained Energy-Based Models (EBMs) is novel and an interesting connection between generative modeling and geometric data analysis. It may be that this specific approach has not been explored previously because the significant computational expense of training an EBM could outweigh the benefits for many common applications. Nonetheless, the two proposed metrics, $G_{E_{\theta}}$ and $G_{1/p_{\theta}}$, are intuitive, well-motivated, and demonstrate impressive performance in the experiments.

**Weaknesses:**
The main weaknesses of the paper relate to the practical limitations and the scope of the contribution.

* **Computational Cost and Practical Applicability:** The proposed method hinges on having a well-trained EBM. Training EBMs, especially on high-dimensional data, is computationally intensive and challenging. This significant upfront cost may limit the method's practical adoption, especially when compared to the much cheaper baseline methods. The paper does not provide strong motivation for a real-world application where the improved geodesic quality would justify the cost of training an EBM from scratch, raising the question of whether the method is primarily intended for scenarios where a suitable EBM already exists.

* **Focus on Conformal Metrics:** The paper's choice to focus on conformal metrics is a key aspect of its scope. This design choice assumes isotropic geometry and cannot capture more complex, anisotropic structures that may be present in the data.

Overall, this is a strong paper with a novel idea that is well-executed. My "Weak Accept" recommendation is based on the fact that while the contribution is elegant and effective, its practical significance is somewhat diminished by the high computational cost and the lack of a clear, compelling application.

---

> ### Author Rebuttal · Authors · 2025-07-29
>
> We sincerely thank the reviewer for the thoughtful and constructive feedback. We particularly appreciate the recognition of our paper’s originality, clarity, and experimental design. Below, we address the two major questions, which both touch on the foundational motivations behind this work. We also answer the additional concerns of the reviewer.
>
>
> * __Q1: On the restriction to conformal metrics:__ This is an excellent and timely question. Our initial experiments were not limited to conformal metrics. We tested the Stein metric $s_\theta(x).s_\theta(x)^T$, s.t. $s_{\theta}(x) = \nabla_x E_{\theta}(x)$, which has been proposed as a valid Riemannian metric for EBMs [25], and also explored the Hessian-based metric in early trials. While both define valid geometries, we found that conformal metrics matched the performance of the Stein metric while offering clearer interpretability and significantly lower computational cost. Furthermore, for the Gaussian case, the Hessian of the energy  yields geodesics that are straight lines—ignoring mass concentration around the mean—which we found undesirable. Finally, computing and storing the full metric tensor incurs a quadratic cost in dimension. __Conformal metrics align with density level sets and produce isotropic, interpretable geodesics. In contrast, non-conformal metrics can introduce anisotropy and directional bias, offering flexibility but at the cost of control and interpretability.__ For these reasons, we chose to focus on the conformal case. To clarify this design choice, we propose adding the following sentence to the conclusion (line 346): _“While more complex non-conformal metrics (e.g., the Stein metric) are accessible from the EBM score, we found that conformal metrics offer comparable performance with simpler interpretation and reduced computational cost, justifying our focus in this work.”_
>
> * __Q2: On envisioned applications and cost-benefit tradeoff:__ We agree that assessing the cost-benefit tradeoff is essential when evaluating practical impact. Training EBMs is indeed computationally expensive, but we believe the cost is justified in domains where (i) data are high-dimensional and sparse, and (ii) precise control over the geometry of the latent space is critical. In these domains, high-fidelity geodesics ensure that interpolated paths (e.g., geodesics) remain meaningful and do not cut through implausible regions. __A prime example (and the original motivation for this work) is the modeling of neural manifolds__: the geometry traced by high-dimensional population neural activity recorded over time. In this context, a low-fidelity approximation of the data geometry is insufficient, as downstream interpretations rely on fine-grained geometric structure. We are exploring this in an ongoing work in which we show that geodesic length in learned neural manifolds correlates with human reaction times in object recognition tasks, suggesting that energy-derived geometry can capture behaviorally relevant structure. __This paper lays the theoretical and empirical foundation for such applications by validating the method on controlled benchmarks.__ To better highlight the cost-benefits tradeoff of our method, we propose adding the following sentence to the discussion (l. 356): _“This approach is particularly relevant for neuroscience, where datasets are high-dimensional, often sparsely sampled, and where understanding the geometry of neural population activity is central to scientific insight. In such settings, high-fidelity geodesics are essential for capturing the true structure of neural trajectories—approximations may distort the manifold and lead to misinterpretation of brain dynamics. While training EBMs is costly, the benefits in terms of interpretability and geometric accuracy make this approach compelling for applications where precision is critical.”_
>
> * __Q3: On geodesic behavior with manifold gaps.__ This is an important and insightful question. When the data manifold contains significant gaps—such as missing intermediate orientations—our method does not impose a fixed strategy but instead lets the learned energy landscape shape the geodesic. Since the Riemannian metric is derived from the EBM and inversely related to data density, geodesics are encouraged to follow high-density (i.e., plausible) regions. __However, if no high-density path exists between two endpoints, the geodesic may cut across a low-density gap if doing so yields lower accumulated energy than detouring through unrelated regions (e.g., another character’s manifold).__ This behavior reflects a key strength of our framework: geodesics adapt to the structure learned by the EBM and naturally trade off between identity preservation and plausibility. In ambiguous cases, the path reflects the most efficient solution under the current energy landscape. We agree that this behavior would be best illustrated with a concrete simulation. While this was beyond the scope of our initial evaluation, __we will include a follow-up analysis in the supplementary material to examine geodesic behavior on discontinuous or sparse manifolds__. This will help clarify when and how shortcuts or detours emerge under different density configurations.
>
> * __Q4: Clarification on Figures 1 and 3:__ Thank you for pointing this out. We will revise the captions to improve clarity. For Figure 1, we will explicitly mention that the trajectories are projected into the first two PCA components of the latent space. For Figure 3, the x-axis represents optimization steps, and the y-axis is the Euclidean norm of the geodesic velocity vector at each step. We will also expand the paragraph describing Figure 3 to clarify the stability behavior and interpret the drop in step size over training.
>
> * __Q5: On finite difference stability:__ We observed that computing velocities via finite differences was more stable and accurate (and also much faster) than using full auto-differentiation. __This is likely due to the fact that finite differences avoid computing second-order gradients through the geodesic interpolant network, which can be noisy or unstable in practice.__ Additionally, the network typically takes the time in input with a time embedding layer (with high frequency sine functions, a la flow matching), whose first order derivative can be ill-conditioned. Finally, finite differences act as a form of local smoothing, making the velocity estimate less sensitive to local curvature irregularities. This contributes to both numerical stability and faster convergence.
>
> * __Q6: On grayscale robustness of plots:__ Thank you for the suggestion. We will update key visualizations to improve grayscale robustness by using colorblind-friendly palettes, distinct line styles, and/or markers to ensure readability when printed in black and white.
>
> We hope these clarifications address your concerns and will contribute to a positive reassessment of the paper. We remain fully open to further discussion and sincerely thank you again for your thoughtful and constructive review.

---

> > ### Comment · Reviewer_z3Pm · 2025-08-01
> >
> > Thank you for your response to my review. The clarifications on the choice of conformal metrics and the envisioned applications in neuroscience are helpful for contextualizing the work. I had a few follow-up questions.
> >
> > - Q1: The practical justification for using conformal metrics makes sense. From a purely theoretical standpoint, however, I am still curious: in what scenarios might a metric with directional biases (that is, anisotropy not derived from data density) be advantageous?
> > - Q2: The neuroscience application provides a compelling case where the method's cost-benefit tradeoff is justified. Are there other domains where you envision this high-fidelity geometric modelling would be similarly critical?
> > - Q3: The model's behaviour of potentially cutting across manifold gaps creates a trade-off between path length and adherence to the data distribution/manifold. Your rebuttal suggests this adaptivity is desirable, but I remain unclear on its benefit. For applications where the goal is to model transformations on the manifold, why would taking a shortcut off the manifold be advantageous? This concern is also related to the geodesic's initialization as a correction to a linear path in the latent space which may not have "natural" linear structure. If a path of lower cost exists through an empty region, the model seems incentivized to take it, which may not be the intended behaviour. For instance, on a spherical manifold with a gap, a path might cut through the interior rather than travel along the surface (depending on say the scaling of $\alpha / \beta$, and it is not obvious why this adaptivity/shortcut would be preferable. I was wondering if you could clarify.
> >
> > A few further technical questions:
> > - How do you guarantee that the metric remains positive-definite? The calibration procedure relies on sampling, but a true guarantee would require knowing the global minimum of the energy function, which is intractable.
> > - Does the finite-difference approximation for computing the path's velocity, while practically stable, introduce any systematic bias into the final geodesic paths?

---

> > > ### Author Response · Authors · 2025-08-04
> > > **Response to follow up questions (1/2)**
> > >
> > > We are grateful for your continued engagement with our work and for raising these deeper theoretical questions. Below, we respond to each of them:
> > >
> > > * **Q1: On the interest of directional biases:** Anisotropic metrics are useful when variability in the data is not uniform across directions. A concrete example comes from robotics: consider a robot arm that learns a latent representation of its 7-DOF joint configurations from demonstration. In this space, some directions correspond to easy, frequent motions (e.g., elbow flexion), while others correspond to mechanically costly ones (e.g., shoulder rotation due to torque limits or joint coupling). A conformal metric treats all directions equally at a given point, regardless of how natural or efficient they are. In contrast, an anisotropic metric can stretch directions that are difficult or undesirable, and compress directions that are efficient or preferred. This results in geodesics that naturally avoid problematic movements and favor safe, efficient ones. In short, anisotropy doesn’t just reflect where the robot has been—it encodes how feasible or desirable it is to move in specific directions, enabling more realistic and reliable path planning in learned latent spaces.
> > >
> > > * **Q2: About applications:** Here also, robotics offers a clear example where our method's cost-benefit tradeoff is justified. In motion planning, inaccurate geodesics may pass through physically unreachable or unsafe configurations. High-fidelity geodesics respect mechanical constraints and result in safer, more reliable movements—especially in learned latent spaces where naive interpolation can be misleading. Medical imaging is another domain where accurate geometry is essential. In radiology or brain imaging, high-dimensional scans (e.g., MRI, CT) are often projected into a latent space to track anatomical changes over time or across patients. When modeling disease progression (e.g., from healthy to pathological states), low-fidelity paths may cut through anatomically implausible regions, distorting the trajectory and potentially leading to incorrect clinical interpretations. High-fidelity geodesics help ensure that interpolated paths reflect realistic, medically valid transformations. In general, high-fidelity geodesics are most important when the latent trajectories are used for downstream interpretation (as in medical imaging, or to model the neural manifold), or when they must adhere to strict physical or mechanical constraints (as in robotics). In such contexts, high fidelity gendesics is essential ensuring meaningful and reliable outcomes. To make this point more explicit, we propose to include these intuitive application domains in the discussion section of the paper.
> > >
> > > * **Q3: On geodesics cutting across manifold gaps:** Thank you for raising this important point. It highlights a central trade-off in our approach: geodesics in our framework are guided by the energy landscape learned by the EBM, not by a predefined manifold. As a result, they tend to follow high-density regions but are not strictly constrained to stay on the empirical data manifold. This behavior is intentional. In domains where data is sparse or incomplete—such as neural or perceptual spaces—forcing paths to remain on the manifold can lead to detours through irrelevant or unrealistic regions. Allowing geodesics to cross low-density gaps when it reduces total energy enables more plausible and efficient transitions. In this sense, the adaptivity is a strength, not a flaw. That said, we agree this flexibility may not be desirable in all settings. For applications requiring strict manifold adherence (e.g., known physical constraints), additional biases can be introduced. For example, the metric can be penalized more strongly in low-density regions by adjusting the $\alpha  \/ \beta$ scaling or adding explicit regularization to discourage off-manifold paths. We also acknowledge the concern about initializing geodesics from straight lines in latent space. This initialization serves only as a starting point, and the optimized path typically deviates significantly. Still, in poorly structured latent spaces, the initialization may influence the result. This could be addressed with alternative strategies, such as sampling multiple random trajectories for initialization. In summary, our method imposes a soft constraint: it prioritizes plausible, low-energy transitions over strict manifold adherence. This flexibility is valuable in many real-world scenarios, but our framework can also support stricter constraints when needed.

---

> > > > ### Author Response · Authors · 2025-08-04
> > > > **Response to follow up questions (2/2)**
> > > >
> > > > * **Q4: Concerning the positive-definiteness of the metric:** The positive-definiteness of our metric is guaranteed by construction, independently of the scaling constants. The energy function $E_\theta$ is strictly positive due to architectural choices in the EBM (final ReLU layer). As a result, both metrics (i.e $G_{E_{\theta}}$ and $G_{1/p_{\theta}}$) are scaled identities with strictly positive scalar factors across the entire space. You're absolutely right that our calibration procedure (used to normalize the metric via \alpha and \beta) relies on sampling rather than knowledge of the global energy minimum. However, this only affects the relative scaling of the metric, not its definiteness. We agree that the positive range of the energy function was not clearly emphasized in the appendix and will clarify this point in the camera-ready version.
> > > >
> > > > * **Q5: On the finite difference approximation:** In practice, we did not observe any significant bias in the trajectory computed with finite difference vs the trajectory computed with full autodifferenciation. We tested several finite difference schemes, including central differences (i.e., the average of forward and backward steps) and midpoint estimates, and found no meaningful differences in the resulting paths. The only parameter that had a notable impact was the number of time steps. Using too few steps led to imprecise velocity estimates that deviated from autodiff-based results. With T=100 steps, the finite difference approximation proved both accurate and stable, making it a reliable and efficient alternative to autodifferentiation.

---

> ### Comment · Reviewer_z3Pm · 2025-08-07
>
> Thank you for your detailed responses.
>
> I agree that including the additional application areas you mentioned would strengthen the paper by better justifying the method's cost-benefit tradeoff. Thank you also for clarifying how positive-definiteness is guaranteed by construction.
>
> Regarding your response to Q3, I am still not fully convinced that allowing geodesics to cross low-density gaps is an inherent advantage. In situations with insufficient data, finding a principled geodesic is fundamentally challenging for any method. The behaviour of your model is a direct consequence of its objective, but whether this outcome is desirable is highly context-dependent. While this is not a flaw of your method (as no approach can perfectly resolve data sparsity), I would be hesitant to frame this specific adaptive behaviour as a definitive strength.
>
> Overall, I believe this is a very nice work, and I will update my score to reflect this positive assessment.

---

> > ### Author Response · Authors · 2025-08-07
> > **Thank you**
> >
> > We’re pleased the clarifications on positive-definiteness and the expanded application scope were useful.
> > Concerning Q3, we’ll tone down our claim that gap-crossing geodesics are a clear benefit and revise the discussion section of the revised manuscript.
> > Thank you for the constructive feedback and improved score.

---

### Official Review · Reviewer_RvU6 · 2025-06-29

**Clarity:** 3
**Significance:** 3
**Originality:** 3
**Rating:** 4
**Confidence:** 5

**Summary:**

This paper proposes a Riemannian metric on the data manifold and demonstrates the geodesics obtained using this metric. They propose to define a conformal metric using energy-based models (EBM). Specifically, the EBM estimates the probability density function of the data, and it is calibrated and used to rescale the identity matrix to get a metric matrix. This metric is then used to compute geodesics by minimizing the length of a neural-network-parametrized curve. The geodesics are evaluated on toy and real data on the closeness to known ground truth and closeness to the data manifold, outperforming model-free baselines.

**Questions:**

1. The metrics are only evaluated indirectly through the geodesics derived from them. Why not directly evaluate the metrics against the ground truth on a known toy manifold with known analytic-form metrics?
2. For the toy dataset, the ground truth geodesic is obtained from a metric of the same form as the proposed metric (isometric, rescaled from the ground truth density). As a result, the evaluation comes down to how accurately the EBM approximates this probability distribution function. How to justify the assumption that the ground truth follows the same restricted form?
3. The geodesics are only evaluated for “closeness to the manifold”, but not for the length minimization, which is a key characteristic and motivation for computing geodesics. Specifically, the evaluation metrics are based on the cumulative probability density, FID, RMSE vs ground truth derived from the density or points in the dataset. Each each of them quantify if the geodesics goes through high density areas, hence close to manifold. While this is an important characteristic and are useful for getting realistic samples, the actual length with respect to the ground truth metric is not evaluated. In the proposed method, the length is minimized with respect to the proposed metric, but how much does it minimize the length of the ground truth geodesic is not evaluated.
4. Could you compare with the simple baseline of the path from Dijkstra’s algorithm? It connects points on the manifold, so it would probably be a strong baseline for the evaluation metrics used in this paper, which quantify how close the points are to the manifold.

**Ethical Concerns:**

["NO or VERY MINOR ethics concerns only"]

**Final Justification:**

I still have the concerns on the theoretical properties of the proposed method on the ideal case, but I agree it is a more minor one after the clarifications on the problem setup. I appreciate the clarifications and the promise to include additional baseline experiments.

**Limitations:**

Yes

**Quality:**

2

**Strengths And Weaknesses:**

# Strengthes:

1. the method draws a novel connection between EBMs and Riemannian metrics
2. The paper is well-written and clear.
3. The proposed method is easy to compute and can be a plug-in replacement for generative methods requiring metrics (such as [24, 38]), although there is a separate challenge to fit a high-quality EBM.
4. The method has shown good performance for deriving geodesics that are close to the manifold, which can have potential for improving real application.

# Weaknesses:

1. There lacks theoretical results that supports the proposed method, only a heuristic for encouraging geodesics along the data manifold (line 157). This means the contribution of the paper might not be sufficient for a neurips paper.
2. The evaluation only covers closeness to the data manifold, but did not cover length minimization, a key aspect of geodesics, or directly compare the metrics. (see questions)
3. The experiment on the toy dataset is not sufficient, Because for this manifold with only one intrinsic dimension, the curve is the geodesic as long as it is on the manifold. However, in a more general case (for example, a 2-sphere) the curve being on the manifold does not imply it has the shortest length. It would be more convincing to also evaluate the curve length on a toy dataset with intrinsic dimension larger than 1. More baselines such as Dijkstra’s algorithm could also be added.

---

> ### Author Rebuttal · Authors · 2025-07-29
>
> We thank the reviewer for the feedback.
>
> * __About the “lack of theoretical support of the proposed method” (weakness 1):__ We would like to clarify that __the notion of a “data manifold” is not an observable ground truth but a modeling assumption introduced by the experimenter. Consequently, there can be no general theoretical guarantee on the optimality of any chosen Riemannian metric, since manifold recovery from finite, noisy data is an inherently underdetermined problem.__ Defining a metric requires imposing structure—just as in Topological Data Analysis, where researchers rely on persistent homology.  Our approach follows a well-established principle [13, 24, 33, 38]: that data density can guide the design of a meaningful Riemannian geometry. We show that this modeling choice leads to empirically useful geodesics that capture structure in the data. The strength of our empirical validation supports the relevance of this choice. As noted by other reviewers: Reviewer yiCX described our evaluation as “convincing,” Reviewer z3pm highlighted the “impressive performance” of our method, and Reviewer RvU6 acknowledged both its novelty and relevance.
>
> * __About the experiment on the toy dataset (weakness 3):__  In the toy experiment (Section 4.2), the data are sampled from a 2D mixtures of Gaussian arranged along a semi-circle. __While we agree that the shortest path lies along a 1D structure (lying on the $\mathcal{S}^1$), we emphasize that our geodesic optimization takes place in the full ambient space ($\mathbb{R}^2$)__, and the initial trajectory is a straight line between the endpoints—thus not aligned with the manifold $\mathcal{S}^1$. The optimization must therefore rely entirely on the learned Riemannian metric to steer the path toward the data manifold, without any explicit constraints enforcing manifold adherence. This setup makes the problem more challenging than optimizing directly on $\mathcal{S}^1$, and demonstrates the method’s ability to recover geodesics that align with the data structure from off-manifold initializations.
>
> * __Q1-2 (benchmarking geodesics):__ As stated previously, the notion of a "data manifold" is not an observable ground truth but a modeling assumption introduced by the experimenter. In this work, we adopt a common and well-supported choice: equipping the ambient space with a density-based Riemannian metric, following prior work [13, 24, 33, 38]. This assumption defines both the manifold’s geometry and the corresponding geodesics. Under this modeling framework, our evaluations are internally consistent. In Section 4.1, __we construct toy datasets with known density functions, which allows us to define a ground-truth metric and geodesics analytically. This setup enables a principled evaluation of geodesic accuracy using RMSE—not merely of density estimation, but of how well the learned metric captures the induced geometry.__ We agree that this modeling assumption could have been more clearly stated in the paper. To address this, we will add the following sentence at line 165: _“Throughout this work, we adopt the common choice of equipping the data space with a density-based Riemannian metric, thereby defining the geometry of the manifold in terms of data concentration.”_
>
> * __Q3 (related to weakness 2):__ Section 4.3 was specifically designed to address this concern. In high-dimensional settings—as in AFHQ (Section 4.4)—it is indeed difficult to obtain ground-truth geodesics. To mitigate this, we introduced __the rotated character setup, where we can define a meaningful proxy for geodesic paths.__ The latent space is regularized using a triplet loss to ensure that small rotations of the same letter are mapped to nearby points. As a result, the shortest path between two rotated versions of the same character corresponds to the smoothest in-plane rotation, which serves as a valid approximation of the true geodesic in this space. The $\gamma^{\star}$-RMSE metric then quantifies how closely our predicted geodesics align with this ground-truth proxy. To clarify this in the manuscript, we propose adding the following sentence at line 273: _“This setup provides a unique middle ground: although the underlying Riemannian metric is unknown, we can treat the smooth in-plane rotation between two instances of the same character as a proxy for the ground-truth geodesic. Thanks to the triplet loss, the latent space is structured so that nearby points correspond to slight rotations of the same character, making the shortest path between two orientations a meaningful approximation of the true geodesic in the task-relevant transformation space.”_
>
> * __Q4 (about the Dijkstra baseline):__ We agree that Dijkstra’s algorithm can serve as a strong baseline in low-dimensional settings, and __we will include it in Section 4.2__. However, it’s important to note that Dijkstra operates on a discretized graph and must rely on the Riemannian metric derived from our EBM to reflect manifold structure. As such, it does not directly assess the quality of the learned metric, but rather the quality of geodesics we compute. __Extending Dijkstra to high-dimensional spaces is computationally infeasible due to the curse of dimensionality.__ To approximate a geodesic with ( $T = 100$) discrete steps, a naive discretization requires evaluating a graph with ( $\mathcal{O}(d^T) $) nodes, where ( $T$) is the number of discretization points per dimension. Even if the intrinsic manifold has dimension ( $d$ ), the search frontier at each step spans a $(d - 1)$-dimensional surface, resulting in memory costs of $ \mathcal{O}((d - 1)^T)$. This makes Dijkstra intractable beyond very low-dimensional settings (see, e.g., [87, 88] for examples in 3D).
>
> We hope this clarifies our methodological choices and reinforces the coherence of our evaluation framework, demonstrating that our work offers a meaningful contribution to the community.

---

> > ### Author Response · Authors · 2025-08-08
> > **Follow up: invitation to discuss before rebuttal ends**
> >
> > We would like to take this last opportunity before the rebuttal period ends to engage in discussion with the reviewer and clarify a potential misunderstanding regarding the evaluation critique (concerning the ground-truth baseline).
> >
> > The review notes that we did not evaluate our trajectories against geodesics on a “ground-truth” data manifold. In practice, no such “true” data manifold — and therefore no “true” metric — exists independently of modeling assumptions. Defining a metric is inherently a design choice; our work follows a common and well-supported approach in the literature, where the geometry is derived from data density. Our evaluation is consistent with this framework, assessing whether the learned metric yields trajectories that capture structure under the chosen model.

---

> > ### Comment · Reviewer_RvU6 · 2025-08-09
> >
> > Thank you for your clarifications and explanations!
> >
> > I agree with the setup of noisy data rather than recovering the metric of the ideal manifold, and I appreciate the additional clarification sentence on the setup to make it clearer.
> >
> > That said, it would still be informative to show how well the model aligns with the ground truth metric in the ideal case, such as on uniform, noiseless points on manifolds with known metrics. Accompanied with some theoretical results on the expected behavior, it could provide a stronger motivation for the specific forms of definition in Equations (6), (7).
> >
> > Thank you for the additional justification on the rotation as a proxy for geodesics.
> >
> > Dijkstra's algorithm, at least on the toy case, can be a complement to the existing baselines for its simplicity and strong performance. For high dimensional setting, dimensionality reduction such as PCA can be a preprocessing step to reduce the computational cost.
> >
> > In summary, I still have the concerns on the theoretical properties of the proposed method on the ideal case, but I agree it is a more minor one after the clarifications on the problem setup. I appreciate the clarifications and  the promise to include additional baseline experiments. I will update my score accordingly.

---

> > > ### Author Response · Authors · 2025-08-09
> > > **Thank you**
> > >
> > > Thank you for your response.
> > > We are pleased the clarifications were helpful and appreciate your constructive suggestions.
> > > As promised in the rebuttal, we will include Dijkstra’s algorithm comparison trajectories in the updated version for the 2D case.

---

> ### Comment · Area_Chair_86UJ · 2025-08-09
>
> Dear Reviewer RvU6,
>
> All reviewers are supposed to engage with author responses to reviews. Please respond to the authors' response directly as well as in our internal discussion. Given the lack of response so far, I may lower the weight of your review in my assessment.
>
> Best,
>
> Your AC

---

### Official Review · Reviewer_UjCm · 2025-06-30

**Clarity:** 4
**Significance:** 2
**Originality:** 2
**Rating:** 5
**Confidence:** 4

**Summary:**

The authors present a method for finding geodesic interpolations in image latent spaces through the use of pretrained energy based models as a proxy for the Riemannian Metric. They present quantitive results on toy datasets that their method matches the true ground truth geodesics better. On higher dimensional data, they provide qualitative results that their method scales better than existing alternatives and produces more realistic samples along the interpolated trajectory

**Questions:**

- Does asserting that $\mathbf{G}(\mathbf{x}) \propto p(\mathbf{x})^{-1} \cdot \mathbf{I}$ assume an a priori independence of the interpolation directions in latent space? Does this assumption match the true data generating process, or can you provide motivation for why it might hold? In otherwords, I believe this is the same as asking, what are the advantages or disadvantages of using a conformal metric parameterization?
- If authors could release code that would greatly improve the strength of their submission.
- For the RMSE metric of Figure 2, it is not stated to which 'ground truth' metric the LAND and RBF Models are compared. How is this decided? Wouldn't it make sense to compare all models with both 'ground truths'?
- Is the use of the triplet loss for the autoencoder of section 4.3 truly necessary? Shouldn't the infintessimal rotations already be the ground truth geodesics of the dataset if the model is trained that way? Second, why is an EBM not directly trained on these simple images? If you are already training an autoencoder, why not train a VAE and use importance sampling to estimate the data density and use this to compute a metric?

**Ethical Concerns:**

["NO or VERY MINOR ethics concerns only"]

**Final Justification:**

The paper is a solid theoretical and empirical exposition of an interesting idea that is relevant to the generative modeling community. It has the potential for significant follow up work. It is novel, precise, and well clearly presented, and therefore demonstrates a valuable contribution to the community. The authors have clarified all my concerns related to scalability, evaluation, and limitations.

**Limitations:**

The limitations section should be included in the main text, not the appendix, in our opinion, especially given the relevance of the limitations which are not discussed elsewhere in the text.

**Quality:**

4

**Strengths And Weaknesses:**

**Strengths:**
- The related work is very thorough and it is appreciated that the authors cite unpublished work (such as that of Perone) which inspired their work.
- The finding that metrics based on energy more effectively capture the curvature of the data manifold is interesting and welcome.
- The idea and mathematics of the paper are well introduced and motivated, and it is an interesting topic for the community to explore.
- The paper is well organized and presented cleanly.


**Weaknesses**
- The motivation for some of the experimental choices remains unclear, for example, the reason for the use of the triplet loss trained encoder in Section 4.3 is not clear. Furthermore, it is not justified why an autoencoder is needed at all, and why the EBM cannot be trained on raw images.
- The intuition behind the use of (3) over alternative interpolants is not clear.
- 'Code released upon acceptance' is not convincing.
- Despite the authors claiming that their approach is scalable, the highest dimensional space they model is 1024, slightly larger than a raw MNIST image.
- The choice 'large scale' dataset that the authors choose to evaluate on in somewhat questionable.
	- For example, why should we assume that the ground truth animal face geometry even has connected paths between these presented images? Wouldn't it make much more sense to try to interpolate something such as natural videos or motion where the 'correct' interpolation is much more apparant?
- The Limitations section is relegated to the *seventh section* of the appendix, when it should be included in the main text.

---

> ### Author Rebuttal · Authors · 2025-07-29
>
> We thank the reviewer for the thoughtful and constructive feedback. We greatly appreciate the recognition of the paper’s originality, clarity, and mathematical formulation. Below, we address each question and concern in detail, while also proposing concrete edits to clarify and strengthen the manuscript.
>
> * __Q1 : About the form of the Riemannian metric:__ Thank you for raising this important question. The conformal metric we use, defined as $G(x) \propto p(x)^{-1}⋅I$, indeed assumes isotropy at each point in space—it scales all directions equally, without modeling directional preferences or anisotropies. While this does not imply full independence of latent factors, it does neglect directional correlations that could be captured by a full, non-conformal metric tensor. This modeling choice is grounded in a long-standing tradition in Riemannian geometry and manifold learning, where density-based conformal metrics are used to contract high-density regions and expand low-density regions (e.g., [13, 24, 33, 38]). __This heuristic offers a compelling balance between expressiveness and interpretability.__ In our early experiments, we also explored non-conformal alternatives,  particularly the metric explored in the blog of Perone [25] ( $s_{\theta}(x).s_{\theta}(x)^{T}$ with $s_{\theta}(x) = \nabla_x E_{\theta}(x)$). However, we found that (i) it offers no performance gain, (ii) it is harder to interpret, and (iii) it is significantly more expensive to compute due to nested gradients. For these reasons, we focused on the conformal case. We propose to clarify this in the conclusion by adding: _“While more complex non-conformal metrics (e.g., the Stein metric) are accessible from the EBM score, we found that conformal metrics offer comparable performance with simpler interpretation and reduced computational cost, justifying our focus in this work.”_
>
> * __Q2: Code release:__  We fully agree that open-sourcing the code improves transparency and reproducibility. Although the codebase was not ready for public release at the time of submission, we have since prepared a clean version. Due to NeurIPS rebuttal guidelines, we are unfortunately not allowed to share any links at this stage, including an anonymous GitHub repository. Nevertheless, __we want to reassure the reviewer of our strong commitment to open science, and we will publicly release the full code upon acceptance.__
>
> * __Q3 : About the RMSE in Fig 2:__ Thank you for this comment. In Figure 2, the RMSE is computed with respect to the ground-truth geodesics defined by a known Riemannian metric, which in turn is derived from the known data density used to generate each synthetic dataset (e.g., $G(x)\propto p(x)^{-1}.I$). All models—including LAND and RBF—are evaluated against this same reference metric to ensure fair and consistent comparison. We will clarify this point in the caption and main text.
>
> * __Q4 : About the latent space (also related to weaknesses 1):__ This is an insightful question. While it is theoretically possible to train an EBM directly in pixel space, doing so is both computationally intensive and geometrically problematic: __Euclidean distances in image space are not perceptually meaningful. For this reason, we project images into a learned latent space where distances are more semantically aligned with task-relevant transformations.__ In the rotated character experiment (Section 4.3), we specifically use a triplet loss to regularize this latent space. __The goal is to ensure that the shortest path between two instances of the same character with different orientations corresponds to a smooth in-plane rotation.__ The triplet loss enforces local structure such that small angular changes are reflected by small latent distances, while large rotations and identity changes are discouraged. This gives us a meaningful approximation of ground-truth geodesics that we have leveraged to benchmarked the different metrics. In general, the use of a latent space makes EBM training significantly easier, as the latent space is lower-dimensional and smoother than pixel space. To clarify this in the manuscript, we propose adding the following sentence at line 273: _“This setup provides a unique middle ground: although the underlying Riemannian metric is unknown, we can treat the smooth in-plane rotation between two instances of the same character as a proxy for the ground-truth geodesic. Thanks to the triplet loss, the latent space is structured so that nearby points correspond to slight rotations of the same character, making the shortest path between two orientations a meaningful approximation of the true geodesic in the task-relevant transformation space.”_
>
> * __A word about scalability of our method:__ We respectfully disagree with the concern that our method lacks scalability. While the latent dimensionality used in our AFHQ experiment is indeed $1024$, this corresponds to a spatial code of size 16×16×4—identical to the pretrained autoencoder used in Stable Diffusion, which is trained on the LAION datasets. __This dimensionality has been shown to capture rich visual features across highly diverse image distributions, indicating that it is not a limiting factor.__ Importantly, our method operates directly on this latent space without architectural changes, demonstrating that it is readily compatible with state-of-the-art, large-scale generative pipelines. Our goal here was not to push to the largest possible models, but to validate the method in a regime already relevant to modern generative pipelines. The method itself imposes no architectural limitations and can easily be scaled to more complex datasets or higher-dimensional latents given appropriate compute resources..
>
> * __Concerning the geodesic interpolant:__ Our geodesic interpolant follows the formulation introduced in prior work [28], which decomposes the trajectory into two components. The first is a fixed linear interpolation, $(1-t)x_0 + t x_1$, which ensures that the curve starts at $x_0$​ and ends at $x_1$​. However, this linear path is only a geodesic in Euclidean space, which does not apply in our setting. To account for the underlying geometry, we add a learnable deformation term, $ 2 t (1-t) \phi(x_0,x_1, t) $, which vanishes at the endpoints and peaks at $t=0.5$, allowing the network $\phi$ to flexibly bend the curve while preserving boundary conditions. This design balances geometric expressiveness with stability and ensures that the interpolant satisfies geodesic constraints in the learned Riemannian space. To give more intuition about the geodesic interpolant, we propose adding the following sentence (l. 142): _"Intuitively, our geodesic interpolant begins with a straight line between the endpoints and uses a neural network to compute a smooth curvature relative to this baseline—bending the path toward regions of higher data density, much like pulling a string taut over a curved surface that reflects the geometry of the data."_
>
> * __About the limitations section:__ We agree with the reviewer. The limitations of our approach (e.g., assumptions about smoothness, dependence on latent space quality) are important and deserve more visibility. We will move the content of the Appendix Section 7 into the main paper and streamline the discussion to make it fit within the page limit.

---

> > ### Comment · Reviewer_UjCm · 2025-08-05
> >
> > We thank the authors for their thorough response.
> >
> > The choice of conformal metric is now understood, and the justification seems fair. The additional clarification to the text is welcomed. We thank the reviewers for their clarification of Figure 2. The scalability of the method is additionally well received.
> >
> > We appreciate the response and we will update our score appropriately.

---

### Official Review · Reviewer_jD3y · 2025-07-01

**Clarity:** 3
**Significance:** 2
**Originality:** 3
**Rating:** 4
**Confidence:** 3

**Summary:**

The paper proposes using Energy-Based Models (EBMs) to derive the Riemannian structure from data. The authors introduce two metrics derived from EBMs based on the energy and likelihood, respectively. Experiments on synthetic datasets with known data density, rotated character images, and high-resolution natural images encoded in latent space show the empirical performance of the method.

**Questions:**

1. How do the authors select the values of $\alpha$ and $\beta$ for the proposed Riemannian metric? The explanation provided around line 191 appears somewhat unclear. Additionally, is the performance sensitive to the choice of $\alpha$ and $\beta$?

**Ethical Concerns:**

["NO or VERY MINOR ethics concerns only"]

**Final Justification:**

Since directly training EBMs on high-dimensional pixel data is a challenging task, this limitation prevents the proposed method from deriving the Riemannian structure directly from pixel data. After the discussions with the authors, I believe that addressing the challenge of training EBMs is not the primary goal of this work. The authors also acknowledge this issue. After thorough consideration, I lean toward accepting the paper.

**Limitations:**

The authors have introduced the limitation of the work in  Appendix G.

**Paper Formatting Concerns:**

I don't find any formatting issues.

**Quality:**

3

**Strengths And Weaknesses:**

## Strength
1. The paper is well structured and easy to follow.
2. The proposed method, which uses an Energy-Based Model (EBM) to induce a Riemannian metric, is reasonable and well motivated.

## Weakness
1. The proposed method appears to rely on a pretrained, high-quality EBM. However, to the best of my knowledge, EBMs are notoriously difficult to train, often suffering from slow convergence and instability. Moreover, designing an effective energy function that balances expressiveness and trainability is a challenging task. Therefore, I am concerned that the proposed method may lack scalability due to its dependence on a high-quality EBM.
2. The experiments seem to be conducted mainly on relatively low-dimensional datasets. The authors have not demonstrated the performance of the proposed method on high-resolution image datasets. Given the issue mentioned above, I am uncertain whether the proposed method would generalize effectively to high-dimensional data.

---

> ### Author Rebuttal · Authors · 2025-07-29
>
> We thank the reviewer for the thoughtful and constructive feedback. Below, we address the main concerns regarding the clarity of our hyperparameter choices, the scalability of our approach, and the stability of EBM training.
>
> * __Q1:__ On the selection of the $\alpha$ and $\beta$ parameters: Thank you for pointing this out. The parameters $\alpha$ and $\beta$ are used to scale the energy (and likelihood) derived Riemannian metrics for comparability across methods. __Specifically, we set $\alpha$ and $\beta$ so that all tested metrics share the same minimum value on the data manifold (normalized to 1) and a maximum value of $10^3$ off the manifold.__ This normalization ensures that observed differences in performance (e.g., FID, smoothness) are not due to scale differences between metrics, but rather to the structure they induce. Empirically, we found that performance was robust to a wide range of $\alpha$ and $\beta$ values, as long as this normalization was preserved. To clarify this in the manuscript, we propose adding the following sentence at line 194: _“We normalize all metrics by choosing $\alpha$ and $\beta$ such that the minimum Riemannian energy on the data manifold is 1, and the maximum energy in off-manifold regions is $10^3$. This allows fair comparison across metric choices without introducing significant sensitivity to hyperparameter tuning. Note that only the ratio alpha/beta influences the geodesics.”_
>
> * __About the training of the EBM:__ __We first emphasize that the core contribution of our paper is primarily methodological, and serves as an additional justification for the development of EBMs.__ Our contribution is not only to propose interpolants while using EBM and metrics, it is also to demonstrate that EBM can be used to define a geometry over the data manifold. Geodesics interpolants are merely a consequence of this modeling choice. We are optimistic that further developments in EBM training will enhance our results (backed by the elements below). We agree that training EBMs can be challenging, particularly in high-dimensional settings. However, recent advances—some of which we leverage in this paper—have made EBM training significantly more stable and scalable:
>    * __Latent space modeling:__ Rather than training EBMs on raw pixels, __we operate in a pretrained latent space that is smoother, lower-dimensional, and semantically structured—greatly improving training stability and efficiency.__ For our high-dimensional experiment (AFHQ), we used the same VAE encoder as Stable Diffusion, which maps natural images into a 1024-dimensional latent space. This latent space has been shown to capture rich, fine-grained structure in large-scale datasets like LAION-5B. As such, the use of a 1024-dimensional latent does not reflect a limitation of our method, but rather a design choice aligned with modern generative modeling practices.
>    * __Denoising Score Matching regularization:__ While not widely documented in the literature, __we found that adding a denoising score matching (DSM) term to the contrastive divergence loss significantly improves training stability and convergence of the EBM.__ This regularization smooths the energy landscape and reduces sensitivity to small-scale noise in the latent space—without compromising the model’s expressiveness. Importantly, it does not impair the key advantage of contrastive divergence: the learned energy landscape remains global (i.e., not tied to a specific noise level), which is essential when using it to define a Riemannian metric. Empirically, this regularization appears to improve both performance and robustness. We are currently preparing a dedicated article that systematically benchmarks its impact across tasks and architectures. To make this point explicit in the manuscript, we propose adding the following sentence at line 185: _“To improve stability, we regularize the contrastive divergence loss with a denoising term, which preserves the global structure of the energy landscape while enhancing convergence—a technique we find both effective and broadly applicable.”_
>    * __Established EBM training techniques:__ In addition to the above, we apply well-known stabilization techniques from the EBM literature [16], including __support clipping__ (to constrain latent codes during Langevin dynamics) and __replay buffers__ (to initialize Langevin sampling from previously generated examples, with periodic resets). These practical tricks further contribute to stable training dynamics.
>
>    Together, __these strategies allow us to train stable and expressive EBMs at scale__—without compromising the quality of the learned energy landscape. We believe that these practical insights will facilitate the adoption of our method in more demanding applications. We thank the reviewer again for these valuable comments, which led us to clarify key aspects of our approach and improve the overall quality of the paper.

---

> > ### Comment · Reviewer_jD3y · 2025-08-05
> >
> > I thank the authors for their responses. The rebuttal partially addresses some of my concerns. However, the authors do not appear to have responded to my concerns regarding the scalability of the proposed method to high-dimensional data (weakness 2). As this concern remains, I will maintain my current score. Could the authors please provide further discussion on this point?

---

> ### Author Response · Authors · 2025-08-05
> **Response to reviewer jD3y**
>
> We thank the reviewer for continuing the discussion and appreciate the opportunity to clarify our response regarding scalability. **There may have been a misunderstanding, as our initial rebuttal already addressed this point in the paragraph titled ‘Latent space modeling’. Nonetheless, we are happy to take this opportunity to further clarify and elaborate on it.**
>
> Our method ensures scalability by operating in pretrained latent spaces, rather than directly training EBMs on high-dimensional pixel data—a task known to be challenging. For the AFHQ dataset (composed of high-resolution natural images of size 512x512), we use the latent space of Stable Diffusion’s (i.e. a pretrained VAE, which is 1024-dimensional). This latent space has been shown to capture rich semantic structure across large-scale, high-resolution datasets such as LAION-5B [18]. **This design choice aligns with standard practice in modern generative modeling and, as demonstrated by latent diffusion models [18], improves the geometric smoothness of the latent space and enhances the scalability of training by reducing the dimensionality and complexity of the data representation.** As a result, it mitigates the typical issues of instability and slow convergence in high-dimensional EBM training and enables our approach to scale effectively. **To further enhance scalability and robustness, we incorporate several stabilization techniques, including denoising score matching (DSM), replay buffers, and support clipping.** These techniques improve training dynamics without compromising model expressiveness. Importantly, our goal in this paper is not to push toward the largest possible models, but to validate the method in a regime that is already aligned with modern generative pipelines. Given sufficient compute resources, our approach can be straightforwardly scaled to more complex datasets or higher-dimensional latent spaces.
>
> We acknowledge the reviewer’s feedback that these points could be emphasized more clearly. We propose highlighting explicitly in the manuscript that operating within pretrained latent spaces is not merely a convenience but a deliberate methodological choice to ensure scalability to high-dimensional, complex datasets.

---

> > ### Comment · Reviewer_jD3y · 2025-08-08
> >
> > Thank you for your detailed response. As mentioned above, my main concern is that the proposed method relies on a well-behaved pretrained EBM. Since directly training EBMs on high-dimensional pixel data is a challenging task, this limitation prevents the proposed method from deriving the Riemannian structure directly from pixel data. I have carefully read the authors’ rebuttal to other reviewers, and I believe that addressing the challenge of training EBMs is not the primary goal of this work. The authors also acknowledge this issue. After thorough consideration, I lean toward accepting the paper and will adjust my score accordingly.

---

### Official Review · Reviewer_yiCX · 2025-07-02

**Clarity:** 4
**Significance:** 3
**Originality:** 4
**Rating:** 5
**Confidence:** 4

**Summary:**

The authors propose to use (unnormalized) energy-based models to create a conformal metric for Riemannian geometry, e.g calculating geodesics (conformal refers to the metric being isotropic). The important observation is that normalization is not needed for computing the metric. Two alternative forms are investigated and compared with two state-of-the-art approaches. The energy based models are generally better at capturing the Riemann structure, quantitatively they show better ability to track high-density regions.  The approach is tested on toy data, digits, and more realistic images

**Questions:**

Question 1: Please discuss limitations e.g. computational effort

Question 2: Please reflect on convergence / learning curves for hyperparameters

Question 3: Could simulated data (computer graphics etc.) be generated? - possibly parametrically, to provide ground truth for more realistic problems?

**Ethical Concerns:**

["NO or VERY MINOR ethics concerns only"]

**Final Justification:**

I enjoyed the reading the paper as reflected in my initial score, the paper provides for significant originality and convincing evaluation. Rebuttal confirmed this impression in my view, I maintain my score.

**Limitations:**

Could not locate a discussion of limitations in the manuscript

**Quality:**

3

**Strengths And Weaknesses:**

Strength 1:  Clear research question; i.e., to use energy based models' unnormalized density estimate for defining a conformal metric.

Strength 2:   Clear and simple development of the innovation (i.e., the idea to use EBMs for definition of metric).

Strength 3:  Convincing experimental evaluation on "toy" data (i.e., the results shown in figures 1-5)

Weakness 1: No ground truth for more realistic data (i.e., figure 6 and table 1).

Weakness 2.  Learning curves for hyperparameter optimization missing (i.e., showing the convergence of hyperparameters as functoin of sample size) .

---

> ### Author Rebuttal · Authors · 2025-07-29
>
> We thank reviewer yiCX for the positive feedback, and we are particularly grateful for the recognition of the clarity of our research question, the novelty of our contribution, and the strength of our experimental evaluation.
>
> * __Q1:__ Our method involves two main stages: (i) training an Energy-Based Model (EBM), and (ii) training a geodesic interpolant network. The primary computational cost lies in stage (i).  __To mitigate the expense of training EBMs in pixel space, we follow a common strategy from generative modeling (e.g., Stable Diffusion) and operate in the latent space of a pretrained autoencoder. This significantly reduces dimensionality (e.g., 1024 for AFHQ) and lowers overhead__. Stage (ii) (i.e. training the geodesic interpolant) is efficient and converges quickly. In particular, the use of finite difference to estimate velocities in Algorithm 1 (instead of full auto differentiation) significantly reduces computational cost while maintaining accuracy. Overall, we found the computational effort of the method to be reasonable and not limiting in practice. To clarify this in the manuscript, we propose adding the following sentence at line 343: _“To keep computational cost manageable, we train the EBM in the latent space of a pretrained autoencoder and use finite differences for geodesic optimization, both of which significantly reduce complexity without compromising performance.”_
>
>
> * __Q2 (related to Weakness 2):__ The main hyperparameters in our method are the energy scaling ratio alpha/beta and the number of discretization steps along the geodesic. We found the model to be highly stable across a wide range of scaling values. __These constants are introduced primarily to ensure fair comparison across metrics by matching their dynamic range; they have negligible impact on convergence speed or geodesic quality.__ Regarding the number of steps, smaller values lead to faster convergence but coarser geodesics, while larger values improve geodesic fidelity at the cost of training time. Our choice of T=100 offers a good trade-off and is consistent with prior work (e.g., [13, 24]). To clarify this in the paper, we will include convergence plots for different values of T in the appendix and add the following sentence at line 230:  _“The scaling constants (\alpha, \beta) are introduced to ensure consistent dynamic range across metrics and have minimal impact on convergence or geodesic quality; the number of discretization steps is chosen as a trade-off between efficiency and accuracy, consistent with prior work.”_
>
>
> * __Q3 (related to Weakness 1):__ We agree that our high-dimensional experiments lack access to ground-truth geodesics. __This is a deliberate choice: unlike prior work that evaluates on synthetic manifolds with known metrics, our aim is to learn the Riemannian metric—via an Energy-Based Model—in complex domains where no closed-form metric exists, such as natural images.__ While this makes evaluation more challenging, it also highlights the originality and applicability of our approach. To partially mitigate this limitation, we include the rotated letter experiment, where the transformation space is well understood. The latent space, shaped by a triplet loss, ensures that nearby points correspond to small in-plane rotations of the same character. As a result, the smoothest rotation between two character poses serves as a meaningful proxy for the geodesic. To clarify this in the manuscript, we propose adding the following sentence at line 273: _“This setup provides a unique middle ground: although the underlying Riemannian metric is unknown, we can treat the smooth in-plane rotation between two instances of the same character as a proxy for the ground-truth geodesic. Thanks to the triplet loss, the latent space is structured so that nearby points correspond to slight rotations of the same character, making the shortest path between two orientations a meaningful approximation of the true geodesic in the task-relevant transformation space.”_
>
> Please note that limitations are stated in appendix G. We sincerely thank reviewer yiCX for the constructive feedback, which helped us improve the clarity and completeness of the manuscript.

---

> > ### Comment · Reviewer_yiCX · 2025-08-05
> > **Fine suggestions!**
> >
> > Thank you for the clarifications and suggested updates to the manuscript!
> >  I really enjoyed reading the paper as reflected in my score, and I will maintain the score.

---

> > > ### Author Response · Authors · 2025-08-05
> > > **Thank you !**
> > >
> > > Thank you for your kind words and for taking the time to engage with our work. We're very happy to hear that you enjoyed reading the paper.

---

### Note · Authors · 2025-08-11

We thank the reviewers for their constructive feedback.
We believe the resulting revisions will substantially enhance the manuscript’s clarity and rigor.

---

### Decision · Program_Chairs · 2025-09-17

**Decision:**

Accept (poster)

**Comment:**

This paper proposes a method for deriving Riemannian metrics from pretrained Energy-Based Models (EBMs) to compute geodesics that follow a data manifold's intrinsic geometry. All reviewers recommended acceptance, highlighting the originality of the approach and the clarity of the manuscript. Initial reviews raised concerns about the scalability of EBM training (jD3y, z3Pm) and the evaluation on complex data without ground truth (yiCX, RvU6). The authors' rebuttal addressed these points by clarifying their use of latent spaces for scalability and by discussing why the lack of ground truth data is not a problem in areas where no closed-form metric exists. I agree with the reviewer consensus and believe this work would be a valuable contribution to the conference. I recommend acceptance.